# On the Analysis of the One-to-Many Mapping in Cross-Modality Text-to-Video Generation with Semantic Spaces

## Abstract

Despite recent advances in text-to-video generation, the role of text and video latent spaces in learning a semantically shared representation remains underexplored. In this cross-modality generation task, most methods rely on conditioning the video generation process by injecting the text representation into it, rather than exploring the implicit shared knowledge between the modalities. However, the feature-based alignment of both modalities is not straightforward, especially for the *one-to-many* mapping scenario in which one text can be mapped to several valid semantically aligned videos, a challenge that generally produces a representation collapse in the alignment phase. In this work, we investigate and give insights into how both modalities cope in a shared semantic space where each modality representation is previously learned in an unsupervised way. We explore this from a latent space learning perspective with a plug-and-play framework that adopts autoencoder-based models that could be used with other representations. We show that the one-to-many case requires different alignment strategies than those commonly used in the literature, which struggle to align both modalities in a semantically shared space.

## 1 Introduction

Cross-modality video generation has recently received a lot of attention due to the impressive performance of recent video generators, making it more difficult to distinguish synthetic from real samples. However, regarding the representation learning aspect of this task, particularly when coupled with joint embedding learning, it remains unclear how both modalities cope in latent space and how feature alignment occurs across different approaches. Recent works (Girdhar et al., 2023; Maiorca et al., 2023; Theodoridis et al., 2020) focus on alignment directions in latent space but employ general approaches that do not explicitly address the *one-to-many* mapping scenario, where one input from a source modality can be mapped to $n$ different and valid outputs in a target modality.

In text-to-video generation, the nature of language enables multiple textual descriptions of a single video scene, while simultaneously, a single text description can correspond to multiple valid visual interpretations. In this context, cross-modality alignment is hindered by the one-to-many mapping problem, as a collapse process is unintentionally encouraged in training. In a general one-to-one case, one input text is trained to be associated with one output video, but in the one-to-many case, one input is associated with several cross-modality outputs. Figure 1 illustrates this case. A generic training pipeline in this scenario encourages poor alignment between the modalities that may cause collapse to the most frequent association, a mean representation of it, or even a random and nearby output in latent space.

Beyond the properties of text and video modalities, current text-to-video generation process video chunks rather than full-length videos, typically ranging from 10 to 256 frames (Jeong et al., 2024; Mittal et al., 2017; Polyak et al., 2024; Tang et al., 2023; Wang et al., 2024b). This approach enables data sets of any size to be split into smaller clips for model training. Although data sets vary in difficulty, even densely-captioned videos that provide text for brief time periods undergo this splitting process, generating multiple video clips associated with the same text and thus creating a one-to-many setting.

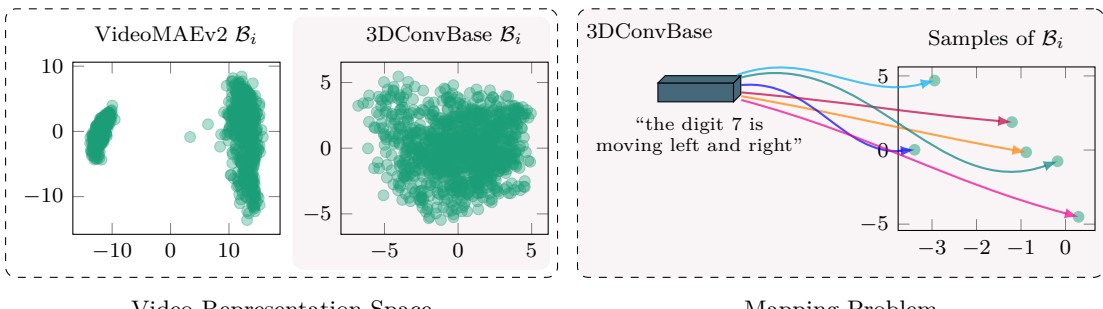

**Figure 1:** The one-to-many mapping case in cross-modality alignment using PCA dimensionality reduction. The leftmost part shows the video representation learning obtained using two models: VideoMAEv2 (Wang et al., 2023) and our 3DConvBase approach, applied to the videos corresponding to the text "the digit 7 is moving left and right" from the SyncDraw-MM (Mittal et al., 2017) data set. The set of semantically aligned videos for a text $i$ is referred to as a *bucket* $\mathcal{B}_i$. The rightmost part shows six random samples from the bucket, illustrating the alignment to be learned in training from one text input to several valid videos in feature-alignment approaches.

Despite being a challenging task, the analysis of the learned joint latent space in the generative context is underexplored. In representation learning, methods rely on classification and retrieval tasks when dealing with a joint embedding approach (Fang et al., 2022; Girdhar et al., 2023; Xue et al., 2023) to validate the learned representation. In text-to-video generation, most methods integrate the text representation through a fusion process within the video generator (Ge et al., 2022; He et al., 2022; Ho et al., 2022; Wang et al., 2024a). In these approaches, the latent representation is held in the background, as this alignment is learned implicitly in the process, with evaluation focusing primarily on video quality.

In this context, works have been proposed to align and generate data from multiple modalities (Tang et al., 2023), where modality-specific models are trained from scratch to learn and regularize a semantically shared representation space. Nevertheless, there are currently several pre-trained models available in the computer vision community for text and video, yet, to the best of our knowledge, few works leverage these pre-trained representations for feature-based cross-modal alignment. Additionally, little is known about this alignment from a latent space perspective, which could provide insight into how the modalities cope in a semantically shared space. In particular, when using autoencoding approaches that regularize the target-modality latent space, analyzing this implicit representation could aid in understanding the alignment process. For the image modality, prior work has explored this relation, from concerns about bias (Gat et al., 2022) when analyzing the latent space, ideal latent distributions for generative models (Hu et al., 2023), to understanding cross-modality alignments for classification tasks (Maiorca et al., 2023), and methods for building joint distributions from autoencoder models (Piening & Chung, 2024; Xu et al., 2019) that enable the generation process.

In this work, we aim for a better understanding of the *one-to-many* case in text-to-video generation. We take the latent space analysis perspective to investigate the structure of the case along with how the modalities cope in pure alignment-based methods. We consider this complementary information with common video quality metrics used for the task, for which we also make an adaptation considering the several videos associated with one input text. We consider a pipeline that leverages models trained in an unsupervised way on their respective modalities, such as text and video encoders, and aligns them in a shared representational space. We show that approaches that directly align these representations (Girdhar et al., 2023) struggle with the one-to-many mapping problem, for which we propose an analysis based on a progressive learning strategy as a baseline. Furthermore, we investigate the impact of self-supervised learning methods originally designed for single-modality representation learning, such as BYOL (Grill et al., 2020), SimSiam (Chen & He, 2021), and VicReg (Bardes et al., 2022), and show their limitations when applied to cross-modal alignment. The main contributions of this work[1] include the following:

- We identify the *one-to-many* mapping scenario as a key challenge in cross-modality text-to-video generation and demonstrate its impact on feature alignment approaches. For our analysis, we adopt a latent

---

[1]The models, checkpoints and data sets generated in this work are publicly available at http://to.be.shared.after.approval.

space perspective to characterize the problem's structure and to explore the relationship between text and video distributions.

- We propose a unidirectional progressive text-to-video model to analyze the *one-to-many* case, mapping text first to a shared semantic space, then to the target video distribution.
- We systematically investigate different mapping functions between the data modalities, showing their impact on the shared semantic space and how the video modality representations can affect alignment.

## 2   Related Works

Text-to-video generation approaches can be divided into those that inject the text as conditioning information in the video generation process and those that aim to learn a generation pipeline by aligning the latent representations of both modalities.

### 2.1   Fusion-based text-to-video generation

In multimodal machine learning, Liang et al. (2024) categorizes fusion into two types: *fusion with abstract modalities* and *with raw modalities*. In text-to-video generation, most methods adopt the former, which considers encoders to represent each modality before applying a fusion method with the two streams of data. The latter employs a fusion process at early representation learning stages, such as using the raw modalities as inputs, and is less explored in the literature.

Text conditioning in fusion-based techniques ranges from simple text injection in the generation process, such as concatenating text and video embeddings (Ge et al., 2022; Wang et al., 2024a), to complex fusion methods based on attention modules or Multi-layer Perceptron (MLP) layers that generate fused text-video embeddings (He et al., 2022; Ho et al., 2022). Both text-only and fused text-video embeddings can be used at different stages or layers of the video generation process. This type of injection aims to ensure that the conditioning information is maintained in the generation and aligned with the desired video semantics. Ge et al. (2022) prepend the text embedding to the video tokens in their transformer-based video generator. Ho et al. (2022) applied MLP layers to the text embedding before adding it to each residual block of their diffusion process. He et al. (2022) concatenates the conditioning information with the latent input with the option to apply or not cross-attention layers before adding it as input to a Latent Diffusion Model (LDM).

### 2.2   Cross-modality generation based on feature-alignment

Unlike approaches that inject the conditioning information into the generation process, multi-modal latent alignment focuses on creating a shared latent space between different modalities. The alignment can support various generation scenarios, ranging from a unidirectional or one-to-one generation (Theodoridis et al., 2020) to any-to-any generation (Tang et al., 2023). In the former, a sample of a target modality $M_1^t$ is generated from an input modality $M_1$, but not the other way around. In the latter, one or multiple target modalities $M_1^t, \ldots, M_m^t$ can be generated from one or multiple input modalities $M_1, \ldots, M_n$.

In the unidirectional context, Theodoridis et al. (2020) proposed an alignment of the latent spaces of two modalities using Variational Auto-Encoders (VAEs) in two separate phases. First, a VAE model for each modality is trained to learn their respective latent spaces. In a second phase, an additional VAE is used to learn a mapping between the two modalities, forming a joint embedding space between them. The alignment is learned by minimizing the Fréchet distance (C. Dowson & V. Landau, 1982) between the distributions and is validated on food image analysis and 3D hand pose estimation. Similarly, in an any-to-any context, CoDi (Tang et al., 2023) employs a two-stage process. The first stage learns the representation of each modality using an LDM. The second stage learns a shared latent space between the modalities, in which one representation is projected onto another by also injecting the target modality in the process, and alignment is achieved through a contrastive approach.

Although these approaches implicitly support text-to-video generation, they did not explore this scenario, particularly the *one-to-many* case. Furthermore, they require joint training from scratch, without leveraging pre-trained representation models for text and video that could benefit the generation process.

Moreover, other methods have been proposed in video-language pre-training for cross-modality tasks such as video-text retrieval and video question answering (Xue et al., 2022; 2023). Xue et al. (2022) proposed a method that encodes high- and low-resolution frames separately before combining them through fusion prior to cross-modal processing. Expanding beyond video-text, Girdhar et al. (2023) proposed a representation-based alignment focused on images as the main binding modality across other modalities, excluding video. In a similar vein, Maiorca et al. (2023) presented a CoDi-like (Tang et al., 2023) approach in the text-image domain, where the decoders for target modalities are pre-trained on their respective source modalities.

Regarding the alignment process, a contrastive approach is generally used, such as InfoNCE (van den Oord et al., 2018), which was adopted in CLIP (Radford et al., 2021). Other works further explore this alignment process (Li et al., 2022; Yeh et al., 2022). DeCLIP (Li et al., 2022) uses a smaller data set of 88M pairs with self-supervised learning applied to both modalities, a multi-view cross-modality loss that extends the multi-crop transformation of Caron et al. (2020), and a nearest-neighbor alignment strategy. Yeh et al. (2022) propose the removal of the negative-positive-coupling effect in learning. Although these works propose different modality augmentations, they are not directly applicable to the *one-to-many* case. Augmenting or changing the text modality could generate multiple valid mappings for a single input, potentially mixing different semantics. Moreover, InfoNCE enforces a one-to-one match between the modalities, treating other valid pairs, such as multiple videos corresponding to the same text in the one-to-many case, as "incorrect pairings" in its batch formulation of positive and negative samples.

Moreover, other works have addressed the one-to-many setting in classification and retrieval tasks (Chun et al., 2021; Khosla et al., 2020; Miech et al., 2020; Patrick et al., 2021; Tian et al., 2023) through: a multi-positive contrastive loss adapted to handle positive-set-based information (Khosla et al., 2020; Miech et al., 2020; Tian et al., 2023); probabilistic-based paradigms (Chun et al., 2021); or external auxiliary components (Chun et al., 2021; Patrick et al., 2021). Since external auxiliary methods introduce additional modules requiring different training setups, the most direct and flexible approach is a contrastive loss tailored to this scenario. However, although these methods offer alternatives, they rely on specific assumptions in their formulation, such as different batch formulations, a maximum number of positive candidates per sample (Miech et al., 2020), or a fixed number of augmentations per sample by external methods (Khosla et al., 2020).

## 3 Unidirectional Progressive Learning for Semantic-Shared Latent Space Alignment

To analyze and understand the one-to-many scenario, we propose a unidirectional approach for cross-modal text-to-video generation as a baseline. Our pipeline consists of two phases: (1) representation learning for each modality, and (2) semantic space alignment based on progressive learning from the text modality. Through this pipeline, we analyze how alignment occurs in two key stages: the shared semantic space between modalities and the target video distribution. First, we assume that video, $v$, and text, $t$, data are in an ideal joint space, $p^*(v, t, z)$, where a latent variable, $z$, holds the joint semantic meaning of them. Hence, our problem is to learn the marginalized distribution $p^*(z)$ given the observations that come from the other marginals that are available at training time. To address the modeling of the semantic space $p^*(z)$, we intend to learn the marginal conditional distributions for each modality. That is, we intend to learn $p(z \mid v) \equiv p^*(z \mid v)$ and $p(z \mid t) \equiv p^*(z \mid t)$. However, learning these posteriors is intractable. Thus, we intend to approximate them with a variational family, $q(z \mid v)$ and $q(z \mid t)$, respectively, parameterized with neural networks. Finally, we need to make them similar so that the semantic information of both is equivalent.

To learn $q(z \mid v)$ and $q(z \mid t)$, we decouple each approximation into two phases. The first phase learns each modality representation, such as $q(z_v \mid v)$ and $q(z_t \mid t)$, and the second phase learns the shared representation of $q(z \mid z_v)$ and $q(z \mid z_t)$, respectively. For $q(z_v \mid v)$, we propose a video extension of a Wasserstein Autoencoder (WAE) (Tolstikhin et al., 2018), trained in an unsupervised manner and, for completeness, is presented in Appendix A. Although a pre-trained model can be incorporated here as well, we select this approach to further evaluate different video architectures in the alignment process. For $q(z \mid t)$, we use a pre-trained text encoder $E_t$, for which we do not impose any restrictions. Figure 2 presents an overview of the pipeline.

**Bridging the Semantic Spaces by Progressive Decoupling.** To learn the shared semantic space given text, $p(z \mid t)$, we approximate this posterior with a parameterized variational family, $q(z \mid t)$, for which

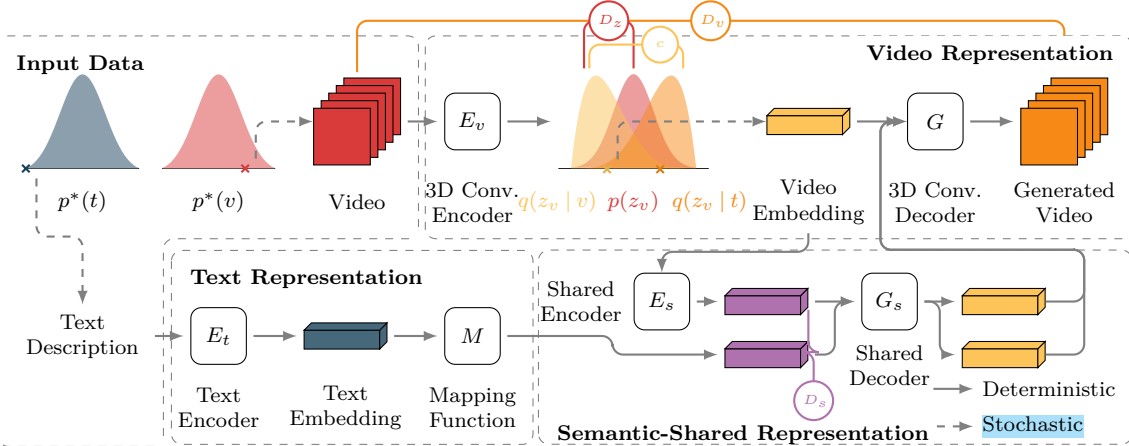

**Figure 2:** Pipeline to learn a joint semantic space $p(z)$ by bridging the gap between the conditional latent distributions of each data modality, $q(z \mid v)$ and $q(z \mid t)$, by minimizing the divergences between shared and target video representations, between the available pairs of text and video. We learn each of these posteriors individually as a variational family using the original data, $p^*(v)$ and $p^*(t)$, through a specific encoder, $E_v$ and $M \circ E_t$, respectively. Given that our evaluation task is video generation from text, we evaluate the quality of the generated videos (by decoder $G$) through a discriminator $D_v$, and by inspecting the similarity on the latent spaces of the decoupling process through additional discriminators $D_z$ and $D_s$.

we propose a two-part decoupling process represented by two models in hierarchical form. The first part encodes the string of words into an embedding $z_t$ using a text encoder model $E_t$. Then, $z_t$ is projected into a shared representation space with a mapping function $M: z_t \to t_s$. We map $t_s$ to the video latent space $q(z \mid t)$, through a generator $G_s$, which is the decoder part of an autoencoder model from the video latent space $q(z_v \mid v)$ to a shared representation between text and video. In this scenario, the encoder $E_s$ generates the shared representation $v_s$ from the video code sampled from $q(z_v \mid v)$.

We found empirically that trying to approximate the target distribution in a one-step approach, i.e. without a hierarchical form, led to the collapse of $q(z \mid t)$ in the *one-to-many* case. The hierarchical latent space decoupling is similar to Xu et al.'s (2019), but we apply it to a modality that is different from the source modality. Moreover, instead of applying a regularization in the latent space in the second stage, our model applies regularization in both intermediate (shared) and target (video) latent spaces.

To link the information between the learned semantic spaces from text, $q(z \mid t)$, and video, $q(z \mid v)$, we use a WAE-based approach similar to the video semantic space, where we define:

$$D_{\mathrm{W}}\left(p^*(z), p(z \mid z_t)\right) = \inf_{q(z \mid z_v), q(z \mid z_t) \in \mathcal{Q}'} \left\{ \mathbb{E}_{z_1 \sim p^*(z)} \mathbb{E}_{z_2 \sim q(z \mid z_t)} \left[c\left(z_1, z_2\right)\right] + \lambda_{z_s} \mathcal{D}_s(q(z), p(z)) \right\}, \qquad (1)$$

such that $\mathcal{Q}'$ is a non-parametric set of deterministic encoders, $z_1 \sim p^*(z)$ is a latent code representing the 'real' distribution, $q(z \mid z_v)$, $z_2 \sim q(z \mid z_t)$ is a generated latent code (through $M$) that depends on text embedding $z_t \sim q(z_t \mid t)$, and $\lambda_{z_s} > 0$ is weight for the divergence measure $\mathcal{D}_s$ between $q(z) = \mathbb{E}_{z \sim p^*(z)} \left[q(z \mid z_t)\right]$ and $p(z)$ representing our shared semantic space. For this phase, we use the cost similarity $c(z_1, z_2)$ as:

$$\begin{aligned} c(z_1, z_2) = \lambda_s \left\|z_1 - z_2\right\|_1 &+ \lambda_{feat}(\left\|G_s(z_1) - G_s(z_2)\right\|_1 + \left\|G_s(z_1) - z_v\right\|_1 + \left\|G_s(z_2) - z_v\right\|_1) \\ &\lambda_{pixel}^s(\left\|G(G_s(z_1)) - v\right\|_1 + \left\|G(G_s(z_2)) - v\right\|_1), \end{aligned} \qquad (2)$$

where $G_s(z_1)$ is the video semantic code from shared code $z_1$; $G(G_s(z_1))$ is the video generated from code $G_s(z_1)$; and $\lambda_s$, $\lambda_{feat}$, and $\lambda_{pixel}^s$ are weights for shared semantic codes, video semantic space, and reconstructed videos terms, respectively.

**Shared Latent Space Divergency.** The divergence measure $\mathcal{D}_s$ is defined as:

$$\mathcal{D}_s(q(z), p(z)) = \mathcal{L}_{D_z^s} + \mathcal{L}_{bucket}, \qquad (3)$$

where $\mathcal{L}_{D_z^s}$ is defined considering a shared semantic space discriminator $D_z^s$ between distribution samples $z_1$ and $z_2$ (similarly to Equation A.5), and $\mathcal{L}_{bucket}$ is a divergence loss based on a bucket approach.

In the *one-to-many* case, the text is represented by the same conditioning information, which is mapped to several semantic-related output videos, named a *bucket*. A bucket $\mathcal{B}_i$ is composed of videos with the same semantics of $t_i \in T$, such as $1 \leq i \leq N$ and $N$ is the number of different text samples. The loss $\mathcal{L}_{bucket}$ is defined with the buckets available in a training batch and follows a contrastive approach between the similarities of intra- (same semantics) and inter-bucket (different semantics) samples. Given $z_v^s \sim p(z)$ and $z_t^s \sim q(z)$, we define the loss as:

$$\mathcal{L}_{bucket}\left(z_t^s, z_v^s\right) = \frac{\lambda_{neg}}{N_t}\left(\sum_{i=1}^{N_t}\sum_{j=1, j \notin \mathcal{B}_i}^{N_v} \frac{S_{ij}}{|\overline{\mathcal{B}_i}|}\right) + \frac{\lambda_{pos}}{N_t}\left(\alpha - \sum_{i=1}^{N_t}\sum_{j=1, j \in \mathcal{B}_i}^{N_v} \lambda_{ij}\frac{S_{ij}}{|\mathcal{B}_i|}\right), \qquad (4)$$

where $S_{ij} = \frac{1}{2}(\cos(z_t^s(i), z_v^s(j)) + 1)$ is the cosine similarity between embeddings $z_t^s(i)$ and $z_v^s(j)$ of the batch, $\lambda_{ij}$ is a weight for the intra-bucket pair, which is set $\lambda_{ij} = 1$ if $i \neq j$ (the sample belongs to the bucket but is not the direct match in the batch) and $\lambda_{ij} = \alpha$ if $i = j$ (direct match of the batch). To determine whether a sample belongs to the same bucket, we compute pairwise cosine similarities between $z_t$ representations in the batch, applying a threshold to assess semantic similarity between samples. Alternative strategies, such as using identical representations, could also be employed. The left term in Equation 4 keeps inter-bucket samples far apart, while the right term encourages intra-bucket samples to be closer. The direct match is reinforced to prevent collapse of mapping $t_i$ to the same $z_t^s$ and to maintain sample diversity.

Unlike InfoNCE loss (van den Oord et al., 2018) and its extensions (Li et al., 2022; Yeh et al., 2022), we do not treat only direct matches as positive samples in the pairwise cosine similarity phase (or diagonal match). Rather, we treat all samples within a bucket as positive samples instead of negative samples by the masking approach in Equation 4. Moreover, the bucket loss used in a one-to-one case resembles commonly used losses for contrastive learning, as each bucket will contain a single video associated with each input text. Equation 1 is the overall loss for the progressive alignment approach, with modality representations already trained.

## 4 Experiments: Case Studies

We first present the main components of our analysis protocol (Section 4.1), then examine the *one-to-many* case from a latent space perspective (Section 4.2). We perform this analysis in four targeted stages. First, we address the general structure of the one-to-many scenario. Second, we isolate the video autoencoder model that forms the target modality representation and examine how different architectures generate different target distributions. Third, we analyze how the shared semantic space between text and video is learned, investigating how different models impact cross-modality alignment. Fourth, we present ablation experiments on the alignment process, including self-supervised methods.

### 4.1 Implementation Details

**Modality Architectures.** We use 3D convolutional deep neural networks for our probabilistic encoder $E_v$, deterministic decoder $G$, and discriminator $D_v$. For the discriminator $D_z$, mapping function $M$, video shared encoder $E_s$, and video semantic space generator $G_s$ we use fully connected networks. For video representation, we consider latent spaces with dimension $d_z$ and isotropic Gaussian prior distributions $p_z = \mathcal{N}(z; 0, \sigma^2 I_{d_z})$. We used different $d_z$ depending on the video architecture, but maintained these values in all corresponding experiments and for all data sets. We did not optimize our model for the choice of $d_z$ on any data set.

We use CLIP (Radford et al., 2021) text encoder with its pre-trained model from the ViT-B/32 version. For the video autoencoder (AE), we consider three architectures for comparison. The first is a 3D convolutional network extended from the 2D DCGAN (Radford et al., 2016) guidelines (3DConv-Base). This network does not use attention modules and residual blocks, although we added skip connections to improve its convergence. The second architecture (UNetLDM) is adapted from Rombach et al. (2022) and is based on latent diffusion. This network is extended with 3D convolutional and transposed operations and includes

residual blocks and attention mechanisms. Beyond that, we also include the VDM (Ho et al., 2022) model based on diffusion to have a baseline comparison for video quality only, as this model was not proposed for representational learning with a posterior reconstruction decoder step.

**Alignment Architectures.** We use the following alignment baselines for comparison: ImageBind (Girdhar et al., 2023), CoDi (Tang et al., 2023), and CLIP (Radford et al., 2021). To ensure fair comparison, we extract only the alignment components from these approaches, isolating the alignment method from the modality representation learning, which varies across methods and data sets. We adapt these methods to text-to-video generation in a pure feature-alignment approach, noting that ImageBind and CLIP were originally designed for classification and retrieval, while CoDi targets multi-modal generation. We therefore compare adapted versions of their strategies focused on the alignment component within our text-to-video generation framework, not using their modality-specific representation architectures. This includes the projection from the video and text semantic spaces to the shared part and the loss function considered for the approach[2].

For ImageBind-based alignment, we adopt its linear projection layer approach, which maps video and text representations to a shared semantic space. The CLIP-based alignment follows a similar approach, including the use of InfoNCE (van den Oord et al., 2018) contrastive loss. For fair comparison with our baseline, we also consider a variant employing our default mapping function architecture rather than a projection layer. Since CoDi (Tang et al., 2023) follows a different paradigm, we implement an alternative inspired by their model. Instead of using representation injection through cross-attention layers in all modality autoencoders, which are based on LDMs, we used the UNetLDM autoencoder with our mapping function architecture, which is based on a non-sharing representational framework, i.e., video embeddings are not passed through the text encoder and vice versa. For all baselines except CoDi, we use the 3DConv-Base video autoencoder. Additionally, all baselines employ InfoNCE (van den Oord et al., 2018) as alignment loss.

**Data Sets and Metrics.** We consider three data sets of increasing complexity that present the one-to-many case: Moving MNIST (Mittal et al., 2017) (SyncDraw-MM), KTH Human Action (Schuldt et al., 2004), and TACoS Multi-level Corpus (Rohrbach et al., 2014). Additionally, for the first case study, we consider Panda70M (Chen et al., 2024), a larger set based on HD-VILA-100M (Xue et al., 2022) with automatic captioning. For all sets, we sampled 16-frame videos with $64 \times 64$ pixels.

Additionally, we consider objective full-reference measures to evaluate video quality, which include: Peak-Signal-to-Noise Ratio (PSNR), Structural Similarity Index Measure (SSIM) (Wang et al., 2004), and perceptual metrics LPIPS (Zhang et al., 2018) and DISTS (Ding et al., 2022). Furthermore, to evaluate the generated video distributions, we consider both Fréchet Video Distance (FVD) and Kernel Video Distance (KVD) (Unterthiner et al., 2018) metrics. For text-to-video evaluation, unlike distribution-based metrics (e.g., FVD and KVD), the full-reference metrics compare against all elements in a *bucket* (the set of videos paired with the same text) rather than a single direct match, since one input text can correspond to multiple semantically aligned videos. Thus, bucket-based metrics compare a generated video to all videos in its corresponding bucket, with the final score being the best match. This approach avoids assuming a single ground-truth video and instead considers the entire bucket.

For the alignment assessment, we consider the Chamfer (CD) and Hausdorff (HD) distances, which are widely used to measure dissimilarity between two sets of points, where the former focuses on average evaluation while the latter on the worst match (more sensitive to outliers). Since these distances may yield misleadingly lower values in collapse scenarios rather than indicating well-aligned sets, we also consider the uniformity measure of Wang & Isola (2020) as an intra-set metric computed independently for each set.

**Latent Space Understanding.** For analysis, we adopt two dimensionality reduction methods: Principal Component Analysis (PCA) and t-SNE (van der Maaten & Hinton, 2008) that are focused on global and local structure preservation, respectively. For completeness, Appendices B and D also include UMAP (McInnes et al., 2018) results, another locally oriented method similar to t-SNE. Although both t-SNE and UMAP are stochastic methods, we use them to interpret the structures captured from the distributions. While different runs can generate different local structures, we focus on overall distribution relationships and complement

---

[2]These components were obtained by the definition of the original works and their official code repositories: `https://github.com/OpenAI/CLIP`, `https://codi-gen.github.io`, and `https://github.com/facebookresearch/imagebind`.

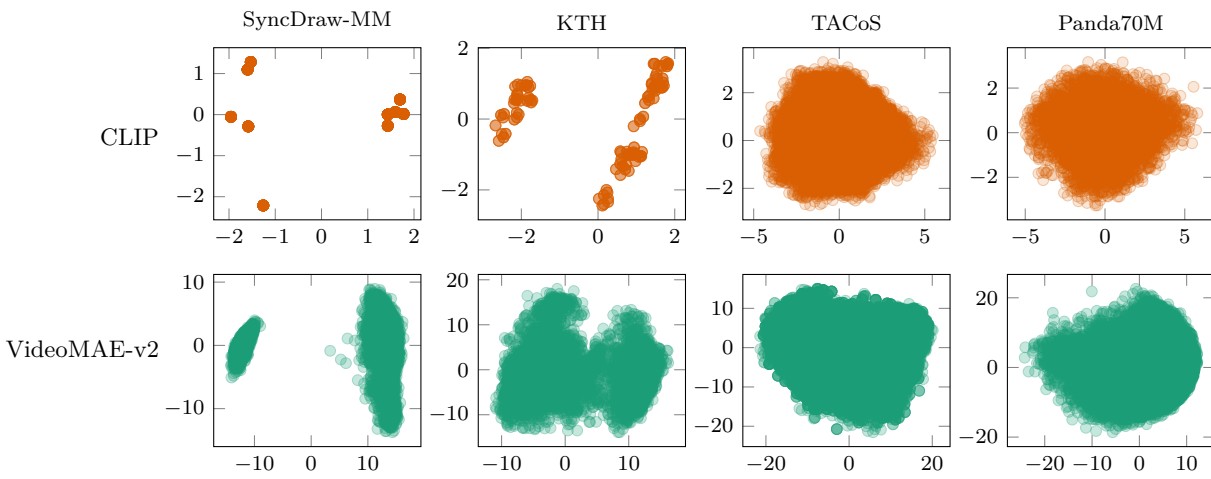

**Figure 3:** Text and video latent spaces generated with CLIP (Radford et al., 2021) and VideoMAE-v2 (Wang et al., 2023) with PCA, respectively (first and second rows), for the training split of each data set, except for Panda70M, where the test set is used due to its already large size.

these local views with global analysis from PCA. In our experiments, multiple runs yielded consistent interpretations, as global distribution relationships remained stable despite local structural variations.

For the video modality, we investigate different autoencoder approaches in Section 4.3. As an unbiased reference outside these autoencoder-based representations for the analysis of the overall structure of the one-to-many case, we use the latent space from ViT-based embeddings obtained using VideoMAE-v2 (Wang et al., 2023), which Ge et al. (2024) demonstrated to be more effective for video representation than models commonly used for FVD computation. Other implementation details can be found in Appendix C.

## 4.2 How is the Latent Space Characterized in the One-to-Many Scenario?

In our initial case study, we examine the general structure of the latent space of the one-to-many case across different data sets including category-oriented, such as SyncDraw-MM, KTH, and TACoS, and an open-domain set such as Panda70M. To understand the impact of individual components on overall alignment, we first isolate each component of the cross-modality process, starting by characterizing its structure. We consider CLIP (Radford et al., 2021) as a pre-trained model for the text modality and VideoMAE-v2 (Wang et al., 2023) as an unbiased video representation outside the models investigated in the second case study.

The visual structure of the one-to-many scenario is shown in Figure 3, with latent spaces from the CLIP text encoder and VideoMAE-v2. Additionally, the largest bucket of each set is shown in Figure 4. In the text spaces, we observe decreasing concentration from SyncDraw-MM, which has fewer, tightly concentrated clusters, to KTH, TACoS, and Panda70M, which show more scattered distributions, with TACoS and Panda70M forming one large, dispersed cluster. Overall, the text modality consists of sparse, concentrated clusters in distinct regions (each representing similar semantics), while the video modality shows dense distributions for videos associated with the same text.

The data sets show decreasing one-to-many difficulty: SyncDraw-MM is most challenging, followed by KTH, with TACoS and Panda70M presenting the lowest difficulty. Although Panda70M is the largest set considered, its one-to-many difficulty level is not lower than that of the TACoS set, which has fewer videos. Considering the ratio of available videos to the number of buckets, Panda70M has more videos per bucket on average than TACoS. Additional information on buckets and video splitting is presented in the Appendix B.

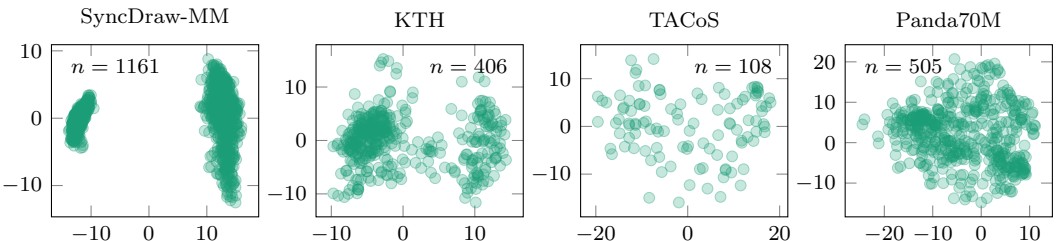

**Figure 4:** Largest bucket for each data set generated with VideoMAE-v2, obtained from the latent space used in Figure 3.

**Table 1:** Quantitative results of video autoencoder (AE) models on the SyncDraw-MM, KTH, and TACoS sets. Best results per column are highlighted. Columns represent the corresponding metric and its values the model results over the test set. Notation: mean over the images ($\pm$ standard deviation), $\uparrow$ indicates that higher is better and $\downarrow$ that lower values are better. PSNR is in decibel scale (dB); SSIM in $[0, 1]$; LPIPS, FVD and KVD in $[0, \infty]$.

| Data Set | Model \ Metrics | PSNR↑ | SSIM↑ | LPIPS↓ | DISTS↓ | FVD↓ | KVD↓ |
|---|---|---|---|---|---|---|---|
| SyncDraw-MM | 3DConv-Base | $19.1 \pm 1.9$ | $0.89 \pm 0.03$ | $0.08 \pm 0.02$ | $0.09 \pm 0.02$ | 2.62 | 0.003 |
| | UNetLDM | $27.8 \pm 2.2$ | $0.97 \pm 0.01$ | $0.02 \pm 0.01$ | $0.03 \pm 0.01$ | 0.27 | 0.0001 |
| | VDM (Ho et al., 2022) | – | – | – | – | 4.15 | 0.006 |
| KTH | 3DConv-Base | $18.8 \pm 2.7$ | $0.41 \pm 0.16$ | $0.14 \pm 0.07$ | $0.24 \pm 0.05$ | 7.77 | 0.007 |
| | UNetLDM | $21.7 \pm 2.6$ | $0.52 \pm 0.19$ | $0.10 \pm 0.07$ | $0.20 \pm 0.06$ | 5.88 | 0.005 |
| | VDM (Ho et al., 2022) | – | – | – | – | 7.70 | 0.009 |
| TACoS | 3DConv-Base | $18.6 \pm 2.3$ | $0.54 \pm 0.07$ | $0.08 \pm 0.03$ | $0.13 \pm 0.02$ | 26.18 | 0.047 |
| | UNetLDM | $18.7 \pm 2.1$ | $0.52 \pm 0.07$ | $0.06 \pm 0.03$ | $0.13 \pm 0.02$ | 19.67 | 0.039 |
| | VDM (Ho et al., 2022) | – | – | – | – | 10.79 | 0.013 |

> **Finding 1:** The one-to-many scenario generally produces a sparser textual space compared to the dense distribution observed in the video domain. Furthermore, different data set types and sizes exhibit this scenario at varying difficulty levels.

### 4.3 Do Video Representations Characterize the Modality Differently in the One-to-Many Scenario?

Although the video modality can be represented using a pre-trained autoencoder, we trained and evaluated multiple architectures to better understand how different video distributions impact the latent space alignment. Table 1 presents the quantitative results for video generation, while qualitative results are provided in Appendix C. Overall, the UNetLDM model obtained the best results across all data sets, except for TACoS, where VDM (Ho et al., 2022) achieved the best performance on FVD and KVD metrics. Notably, the UNetLDM model, which uses a backbone adapted from an LDM, generates satisfactory results without requiring the diffusion process. In contrast, while 3DConv-Base does not achieve optimal video quality, it produces satisfactory results with occasional reconstruction errors, such as confusing digits 1 and 7.

Figure 5 shows the latent spaces obtained using 3DConv-Base and UNetLDM. Although both models generate high-quality reconstructions, they exhibit different latent space distributions, with UNetLDM producing a sparser space than 3DConv-Base. Examining each space individually reveals different clustering structures, indicating that the models learn different representations for the same task despite identical training parameters and sets. The impact of these differences on alignment is explored in the next case study.

> **Finding 2:** The video representation is affected by its corresponding architecture producing different distributions in latent space.

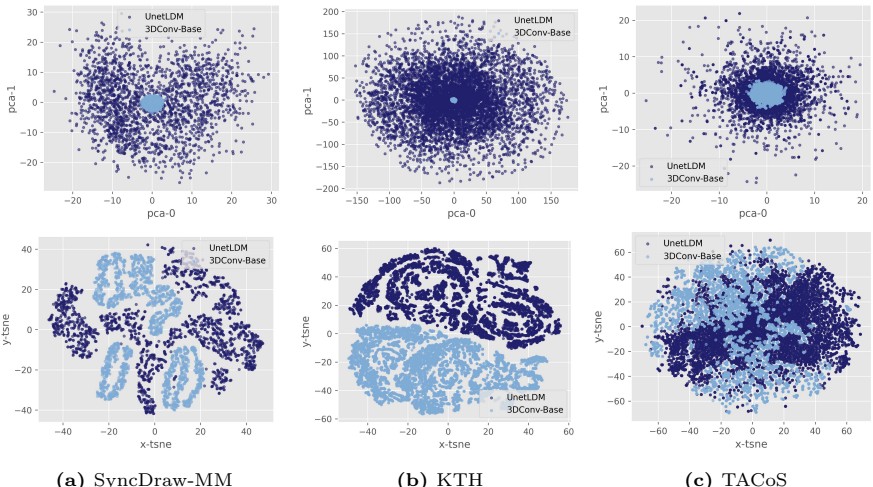

**(a)** SyncDraw-MM     **(b)** KTH     **(c)** TACoS

**Figure 5:** Video semantic spaces from 3DConv-Base and UNetLDM on the SyncDraw-MM, KTH, and TACoS data sets (PCA on first row and t-SNE on second row). For UNetLDM, we first apply PCA to reduce the dimensionality to match the latent dimension of 3DConv-Base.

**Table 2:** Video quantitative results of the feature alignment between text and video modalities on three data sets: SyncDraw-MM, KTH, and TACoS. Best results per column are highlighted. Notation: "B" indicates bucket approach for the metric.

| Data Set | Model \ Metrics | B-PSNR↑ | B-SSIM↑ | B-LPIPS↓ | B-DISTS↓ | FVD↓ | KVD↓ |
|---|---|---|---|---|---|---|---|
| SyncDraw-MM | CLIP-based | $19.7 \pm 1.7$ | $0.852 \pm 0.019$ | $0.23 \pm 0.06$ | $0.14 \pm 0.02$ | 7.48 | 0.008 |
| | CoDi-based | $21.2 \pm 1.6$ | $0.891 \pm 0.014$ | $0.38 \pm 0.03$ | $0.15 \pm 0.02$ | 8.42 | 0.011 |
| | ImageBind-based | $19.2 \pm 1.3$ | $0.849 \pm 0.020$ | $0.25 \pm 0.07$ | $0.14 \pm 0.02$ | 7.73 | 0.007 |
| | Our method - 3DConvBase | $19.6 \pm 1.4$ | $0.855 \pm 0.019$ | $0.21 \pm 0.05$ | $0.14 \pm 0.02$ | 6.77 | 0.006 |
| | Our method - UNetLDM | $21.2 \pm 1.6$ | $0.889 \pm 0.014$ | $0.37 \pm 0.03$ | $0.15 \pm 0.02$ | 8.18 | 0.010 |
| KTH | CLIP-based | $18.8 \pm 3.7$ | $0.27 \pm 0.17$ | $0.27 \pm 0.08$ | $0.30 \pm 0.07$ | 24.09 | 0.037 |
| | CoDi-based | $18.4 \pm 3.2$ | $0.34 \pm 0.19$ | $0.37 \pm 0.08$ | $0.34 \pm 0.09$ | 33.42 | 0.051 |
| | ImageBind-based | $21.0 \pm 2.9$ | $0.35 \pm 0.15$ | $0.26 \pm 0.07$ | $0.30 \pm 0.07$ | 26.92 | 0.025 |
| | Our method - 3DConvBase | $20.7 \pm 2.2$ | $0.358 \pm 0.088$ | $0.26 \pm 0.05$ | $0.31 \pm 0.04$ | 27.51 | 0.028 |
| | Our method - UNetLDM | $19.8 \pm 4.2$ | $0.28 \pm 0.15$ | $0.24 \pm 0.09$ | $0.30 \pm 0.08$ | 23.58 | 0.015 |
| TACoS | CLIP-based | $15.8 \pm 1.4$ | $0.361 \pm 0.064$ | $0.26 \pm 0.07$ | $0.23 \pm 0.03$ | 34.24 | 0.066 |
| | CoDi-based | $16.4 \pm 1.9$ | $0.368 \pm 0.069$ | $0.34 \pm 0.08$ | $0.25 \pm 0.04$ | 37.87 | 0.074 |
| | ImageBind-based | $17.1 \pm 1.9$ | $0.480 \pm 0.069$ | $0.24 \pm 0.05$ | $0.20 \pm 0.04$ | 33.84 | 0.062 |
| | Our method - 3DConvBase | $17.9 \pm 2.3$ | $0.505 \pm 0.083$ | $0.14 \pm 0.08$ | $0.16 \pm 0.04$ | 27.57 | 0.050 |
| | Our method - UNetLDM | $17.3 \pm 2.1$ | $0.504 \pm 0.079$ | $0.29 \pm 0.10$ | $0.22 \pm 0.05$ | 58.88 | 0.103 |

### 4.4 Are There Key Aspects for Learning a Shared Semantic Representation in the One-to-Many Scenario?

We analyze the alignment of text and video modalities using our baseline method from Section 3. For video representation, we use the models evaluated in Section 4.3; for text, we use CLIP text encoder. The alignment baselines include those described in Section 4.1.

Tables 2 and 3 presents the video and alignment quantitative results for these methods, and Figures 6 and 7 show the target video and shared semantic latent spaces obtained with them. In this section, we focus on latent space understanding, while qualitative results for these models are provided in Appendix D.2. Although the alignment approaches differ in architecture and loss function, they produce similar video quantitative results. No specific loss or architecture (projection layer versus our mapping function architecture) appears to confer a significant advantage, and the proposed baseline approach appears to be among the best-performing models. Furthermore, the optimal model varies across data sets depending on their

**Table 3:** Quantitative alignment results of the shared and video semantic spaces for the progressive approaches on data sets: SyncDraw-MM, KTH, and TACoS. Notation: CD in $[0, 4]$; HD in $[0, 2]$; $U_a$ and $U_b$ in $[-\infty, 0]$ correspond to the uniformity of the target and predicted set, respectively.

| | Semantic space | Shared-space | | | | Video-space | | | |
|---|---|---|---|---|---|---|---|---|---|
| Data Set | Model \ Metrics | CD↓ | HD↓ | $U_a$↓ | $U_b$↓ | CD↓ | HD↓ | $U_a$↓ | $U_b$↓ |
| SyncDraw-MM | CLIP-based | 1.25 | 1.272 | −2.53 | −1.40 | 2.73 | 1.540 | −3.27 | −1.69 |
| | CoDi-based | 1.32 | 1.220 | −2.30 | −1.72 | 3.05 | 1.501 | −3.72 | −1.66 |
| | ImageBind-based | 1.26 | 1.166 | −2.54 | −1.69 | 2.68 | 1.452 | −3.27 | −1.80 |
| | Our method - 3DConvBase | 0.01 | 0.578 | −0.72 | −0.69 | 2.10 | 1.333 | −3.27 | −0.70 |
| | Our method - UNetLDM | 0.01 | 0.389 | −0.72 | −0.69 | 2.58 | 1.538 | −3.72 | −0.91 |
| KTH | CLIP-based | 1.12 | 1.333 | −2.38 | −2.13 | 2.84 | 1.523 | −3.46 | −1.61 |
| | CoDi-based | 2.09 | 1.434 | −2.89 | −1.95 | 3.02 | 1.425 | −3.45 | −2.04 |
| | ImageBind-based | 1.78 | 1.339 | −2.67 | −2.04 | 2.80 | 1.450 | −3.46 | −2.16 |
| | Our method - 3DConvBase | 0.13 | 0.879 | −0.47 | −0.44 | 2.53 | 1.389 | −3.46 | −1.51 |
| | Our method - UNetLDM | 0.01 | 0.253 | −0.04 | −0.02 | 1.87 | 1.480 | −3.45 | −1.43 |
| TACoS | CLIP-based | 1.27 | 1.098 | −3.68 | −3.57 | 2.44 | 1.288 | −3.69 | −3.1 |
| | CoDi-based | 1.99 | 1.129 | −3.50 | −3.34 | 2.86 | 1.307 | −3.48 | −3.38 |
| | ImageBind-based | 2.06 | 1.144 | −3.56 | −3.51 | 2.36 | 1.314 | −3.69 | −3.28 |
| | Our method - 3DConvBase | 0.05 | 0.745 | −0.88 | −0.86 | 0.78 | 1.118 | −3.69 | −3.45 |
| | Our method - UNetLDM | 0.0 | 0.005 | 0.0 | 0.0 | 2.00 | 1.468 | −3.48 | 0.0 |

one-to-many complexity. However, latent space analysis reveals important differences that are not apparent in the video quantitative metrics.

**Video Architectures on Alignment.** Although the UNetLDM model provides one of the best results in video reconstruction, alignment using UNetLDM does not produce the best text-to-video results. Note that both our UNetLDM-based method and the CoDi-based alternative employ UNetLDM. Quantitatively, our baseline, despite using different video latent space distributions, yields slightly superior or equivalent video results to the CoDi-based alternative. However, CoDi-based method presents more well-spread distributions although more misaligned spaces compared to UNetLDM (higher CD and smaller predicted uniformity $U_b$ values). We would expect a different alignment pattern between the video models, since the latent space of UNetLDM is sparser than that of 3DConv-Base, as shown in Section 4.3. However, the video representation model appears to play only a partial role in this result, as the UNetLDM-based method generally shows better video alignment than CoDi, except for TACoS, for which neither method achieves proper alignment, with the UNetLDM-based method presenting a collapse from the shared (Figure 7(d)) to video spaces (Figure 6(d)) with CD and uniformity values equal to 0.

Regarding the video spaces produced by UNetLDM and 3DConv-Base, both show a concentration of the generated video codes in regions that do not align with the expected distribution. For the UNetLDM-based method on the TACoS set, even the distribution obtained with the video shared autoencoder - which maps the video latent codes to the shared semantic space and back - does not align with the true distribution (blue color). This suggests that UNetLDM has difficulties with alignment, specially for TACoS, which is partially associated with its architecture and the resulting sparser video space, as different losses and projection architectures are considered with this model which generate more well-spread distributions although still coping with a level of misalignment.

**Projection versus Progressive Architecture.** We observe that the progressive architecture (i.e., our mapping function architecture that is not composed of a single linear projection) generates video latent spaces with less misalignment (smaller CD and HD values) compared with projection layers, although it does not fully prevent a level of collapse on its own (smaller uniformity values when comparing with target distribution). Figure 6 shows more small clusters in the ImageBind and CoDi-based methods, in addition to the concentrated regions of the predicted video codes. In contrast, the CLIP-based alignment, which uses the progressive architecture but shares the same loss function as the other two models (ImageBind

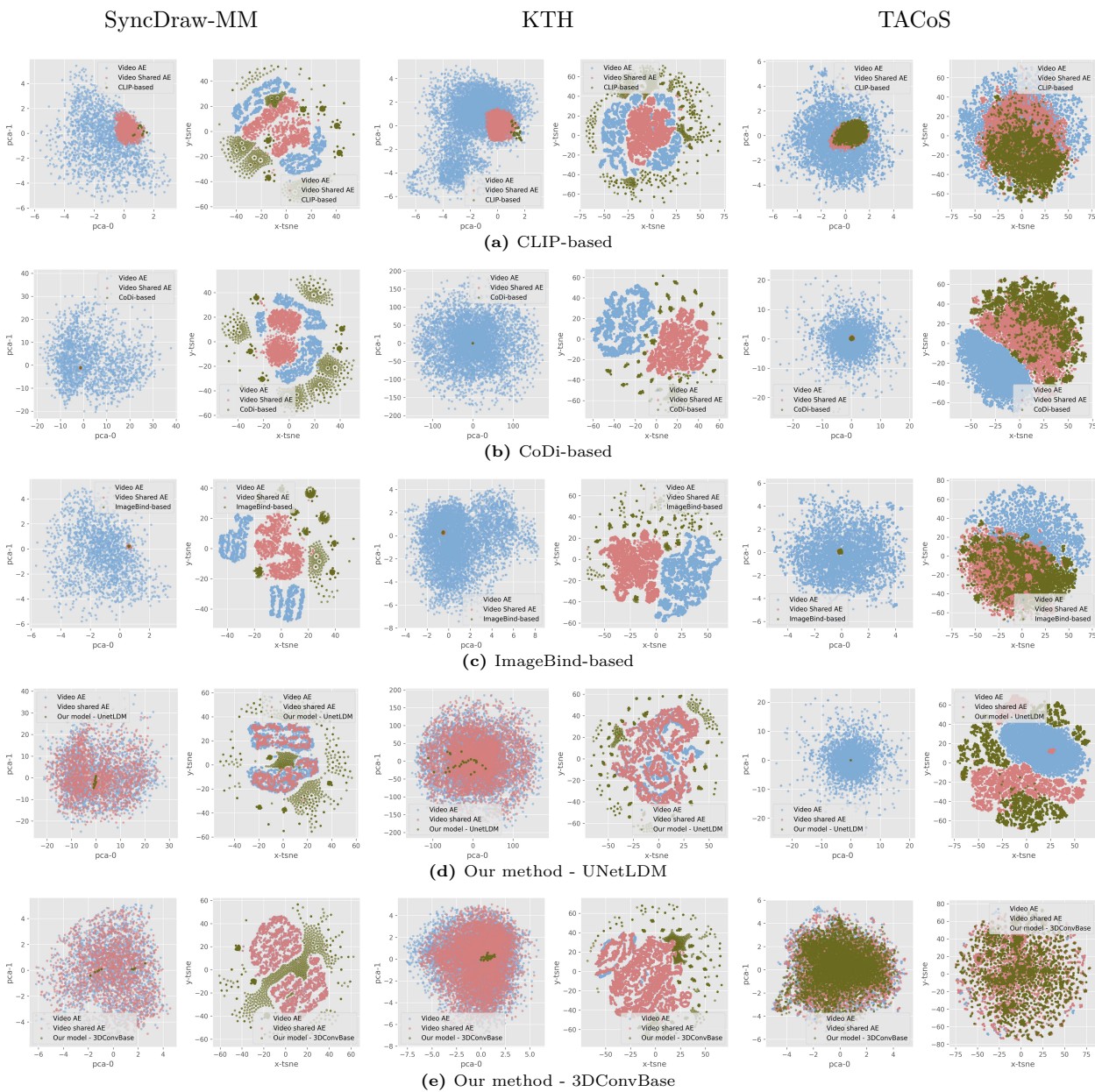

**Figure 6:** Video latent spaces from the feature alignment methods, showing embeddings from the mapping functions, video shared autoencoders, and video representation learning. Results for SyncDraw-MM (columns 1-2), KTH (3-4), and TACoS (5-6) sets using PCA (odd columns) and t-SNE (even columns).

and CoDi-based), shows better alignment although diversity (higher spread) appears more prominent on the projection-layer-based methods.

Additionally, CLIP-based, ImageBind-based, and CoDi-based methods show misalignment of the video shared codes (pink) with the expected distribution (blue) across all sets (Figures 6(a)-6(c)). This misalignment occurs to some degree in other methods as well, e.g., UNetLDM on TACoS. When analyzing the shared semantic space in Figure 7, we observe misalignment in the global structure (e.g. PCA) of the video shared (purple) and predicted codes (green), with the predicted codes lying mostly outside the expected distribution on SyncDraw-MM and KTH. For ImageBind and CoDi-based methods, misalignment also appears in the local structure on the TACoS set, which has lower *one-to-many* complexity. This is further supported

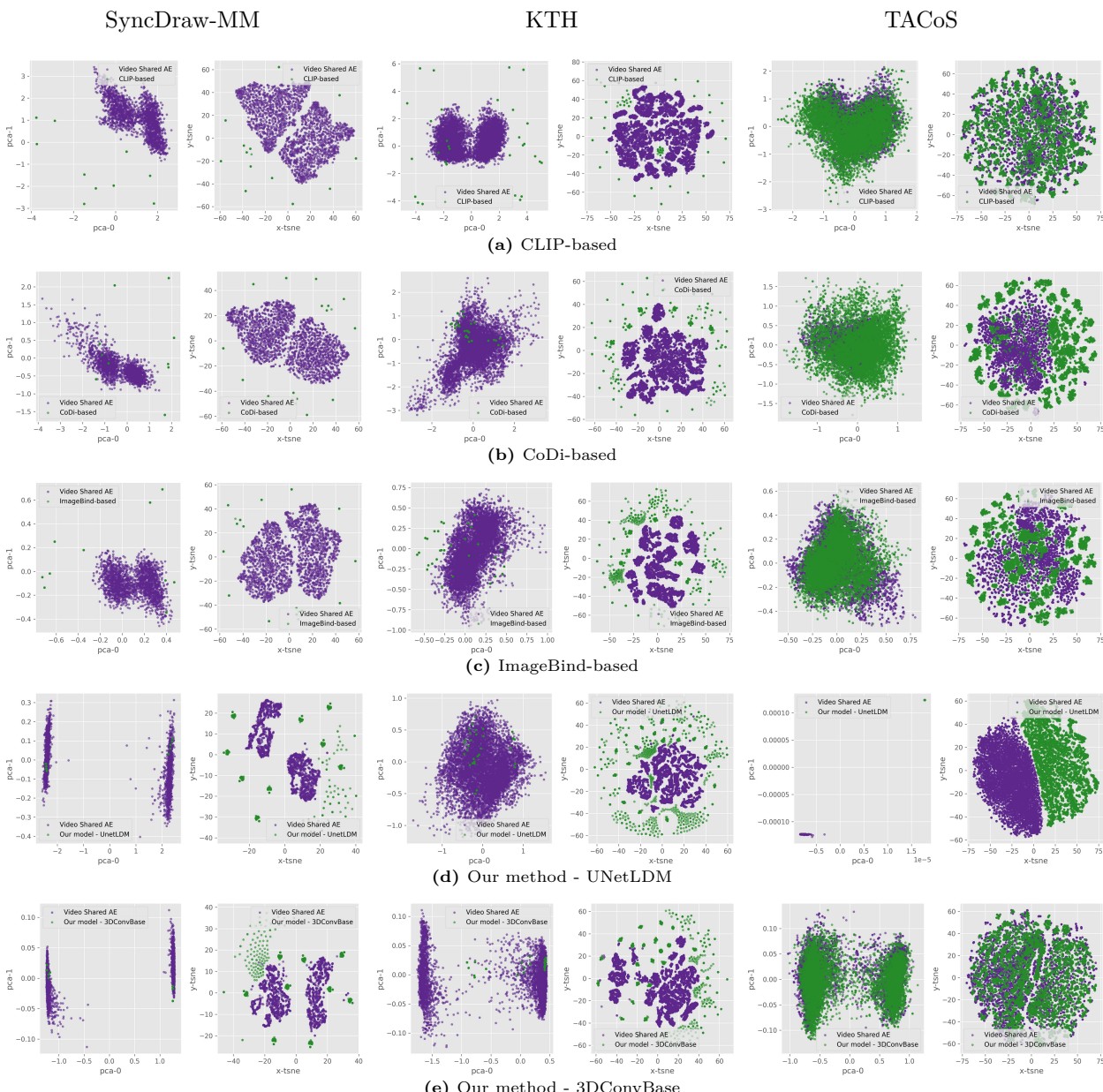

**Figure 7:** Shared semantic latent spaces from the feature alignment methods, showing embeddings from the mapping functions and video shared autoencoders. Results for SyncDraw-MM (columns 1-2), KTH (3-4), and TACoS (5-6) sets using PCA (odd columns) and t-SNE (even columns).

by the shared space alignment results in Table 3, where projection-based methods exhibit higher misalignment (higher CD and HD values) but also more well-spread distributions (uniformity values), whereas our baseline models produce more compact spaces, as reflected by the target uniformity and lower alignment metrics, potentially indicating some degree of collapse. Nevertheless, this appears to benefit the target video space, where progressive approaches achieve better alignment.

**Progressive Alignment Approach.** While the video codes produced by the video shared autoencoder are correctly mapped to the video latent space for our baseline (except for UNetLDM on TACoS), this does not occur entirely for the embeddings coming from the text modality (green), especially at higher task difficulty levels. This may indicate that aligning the video latent space with the shared semantic space is

more straightforward than mapping the text to the target video space. Note that in the video context, we still consider a *one-to-one* mapping, and only from the text modality this becomes our target problem.

Analyzing the shared semantic spaces in Figure 7, we observe poor alignment that propagates to the video latent space. For instance, the UNetLDM-based method incorrectly produces well-separated distributions for TACoS, which from the CD and HD alignment and uniformity values indicates a collapse, while ImageBind-based and CoDi-based methods also exhibit poor alignment (higher CD and HD metrics) in local structure for this set. Regarding the CLIP-based and our baseline methods, we observe better alignment compared to ImageBind and CoDi-based methods. However, when analyzing their video latent spaces in Figure 6, the CLIP-based method shows poor alignment between the video codes from the shared autoencoder (pink) and the expected distribution (blue) for all sets. Our baseline method, although showing better video space alignment than the CLIP-based method, still exhibits a cluster concentration of the generated codes in the video latent space, which is also indicated by the higher uniformity values. Furthermore, its shared space present better alignment metrics that coupled with the uniformity values suggest some degree of compactness (since $U_a$ is close to $U_b$) or collapse between the generated and expected distributions.

These results also highlight a broader limitation: the video metrics used, whether bucket-based or distribution-based, are closely related to video quality. A text mapped to a video code $\hat{v}_1$ could be closer to an expected true code $v_1^t$, yet not close enough for the decoder to properly generate the expected video. A key bottleneck is the decoder's limited capacity to decode from regions near but not within the true distributions, leading to attempted decoding from points unknown to the decoder. These cases appear to generate partially correct videos that are mostly incomplete or missing information. Thus, the metrics may not properly capture the incorrect generation, as similar incorrect decoding can generate similar metric results. Moreover, even latent space analysis, which offers an alternative perspective with valuable insights into alignment, is limited to overall distribution analysis, manual inspection of small regions, and proper metrics for evaluation.

> **Finding 3:** The 3DConv-Base video representation tends to yield better alignment than UNetLDM across evaluated data sets, suggesting that the choice of video representation plays an important role in the alignment process. The progressive architecture generally appears to favor video alignment more consistently across the sets than using linear projection layers. Additionally, progressive alignment tends to enable better alignment between the shared codes derived from the video space and the expected video distribution, suggesting that this mapping may be more easily learned than that from the text modality, which presents the one-to-many case. Furthermore, our baseline loss appears to promote better alignment than InfoNCE, though neither achieves optimal alignment. InfoNCE presents a more dispersed distribution yet fails to properly align even the video shared autoencoder distribution with the true distribution.

## 4.5 Ablation experiments

We conduct ablation experiments to assess the impact of our progressive approach and loss function on the alignment process. Additional studies on video representation learning are included in Appendix D.1. In this section, we use the SyncDraw-MM set, where the one-to-many scenario is more prominent, and the 3DConv-Base as our video baseline. We evaluate a non-progressive method that directly aligns text with video representation without an intermediary step (shared semantic space), using both our baseline loss, InfoNCE (van den Oord et al., 2018), and the StableRep (Tian et al., 2023) multi-positive contrastive loss. We also evaluate our progressive approach adapted to use self-supervised techniques such as VicReg (Bardes et al., 2022) and BYOL (Grill et al., 2020). We adapt these techniques by treating each augmented version as a separate modality (e.g., text and video streams rather than two views of the same modality). Since SimSiam (Chen & He, 2021) produces similar results to BYOL, we only report BYOL. Tables 4 and 5 present the video quality and alignment quantitative results, respectively. Figures 8 and 9 show the latent space analysis, and qualitative results are provided in Appendix D.2.

When comparing non-progressive and progressive approaches, full-reference metrics reveal little difference in generated video quality, while distribution-based results tend to favor the progressive method in most cases. Analysis of the learned latent spaces shows that the non-progressive mapping generates condensed clusters in regions outside the expected distributions and exhibits greater misalignment between modalities in the video space compared to the progressive mapping, also supported by the alignment metrics. Although misalignment persists, generated and expected video distributions tend to show greater proximity in progressive methods than in non-progressive methods, which generally produce more distant global alignments.

Mapping text to video in two phases introduces an intermediate representation (shared semantic space) that is regularized during training, unlike one-phase mapping. This regularization is supposed to enforce alignment in the intermediate phase before generating the target distribution in the subsequent step. In contrast, one-phase alignment could be more challenging as it must generate the desired distribution in a single step, with the model more constrained in performing the transformation between latent spaces. However, we observe that the shared space does not necessarily present well-spread and aligned distributions, with the exception of the progressive model with StableRep, as indicated by the lower CD and uniformity values, which potentially suggest some degree of collapse. Even considering that the target uniformity ($U_a$) is close to the predicted uniformity ($U_b$), this may indicate a certain level of collapse in the target distribution as well, suggesting that the shared space may not be a reliable evaluation target. These results indicate that the shared space acts as a training signal that benefits video space alignment, which shows improvement over the non-progressive approach.

The higher misalignment observed in non-progressive methods is not resolved by the bucket loss alone (Section 3), with both variants yielding similar video distribution- and bucket-based metrics. However, the non-progressive approach shows poor uniformity in the predicted video space ($U_b$) compared to the progressive counterpart. The global structures of non-progressive methods exhibit different relationships and video distribution arrangements in t-SNE and PCA. PCA reveals that generated distributions fall outside the expected regions for non-progressive approaches using InfoNCE and StableRep, also confirmed by higher CD and HD values, and at the same time present well spread distributions (higher $U_b$). This indicates that contrastive losses improve representation quality but hurt alignment. When using the bucket loss, which treats all samples from the same bucket (sharing the same input semantics) as positive samples, the generated distributions exhibit different behavior, where misalignment is lower but persists and predicted uniformity indicates a level of collapse. The bucket loss gives greater weight to the direct match to enforce diversity within the bucket, but this diversity is not achieved with the non-progressive approach.

Regarding the StableRep loss, although yielding slightly better video quality metrics in both progressive and non-progressive versions, we observe a misalignment in the video semantic space between the video autoencoder embeddings and the video shared embeddings of the progressive approach, unlike other approaches which show better alignment. The non-progressive version shows alignment results similar to those of the progressive version. Although, the progressive approach shows a well-distributed shared space in both sides, this does not indicate a significant video space alignment benefit over non-progressive in this case. Moreover, non-progressive with StableRep shows alignment results similar to InfoNCE, although StableRep produces a more spread video distribution, suggesting that the multi-positive setup of this loss may not be well suited for non-progressive approaches such as the one evaluated here.

Considering self-supervised approaches, VicReg and BYOL both generate similar — though not identical — structures in the video and shared semantic latent spaces, with VicReg achieving slightly better video quantitative performance. We observe that their video space distributions are similar to our baseline model (Figure 6(e)), though their differ in the shared space alignment where they present a higher level of collapse. Another related factor is that VicReg and BYOL produce more compact distributions that span narrower ranges along both axes. Furthermore, all evaluated approaches demonstrate the difficulty of aligning text and video on SyncDraw-MM and reveal that these strategies only partially address the one-to-many mapping challenge.

**Table 4:** Quantitative results for ablation experiments on text-video feature alignment using the SyncDraw-MM data set and the 3DConv-Base video autoencoder.

| Model/Metrics | B-PSNR↑ | B-SSIM↑ | B-LPIPS↓ | B-DISTS↓ | FVD↓ | KVD↓ |
|---|---|---|---|---|---|---|
| Non-progressive | $19.7 \pm 0.9$ | $0.856 \pm 0.014$ | $0.20 \pm 0.01$ | $0.13 \pm 0.01$ | 8.47 | 0.011 |
| Non-progressive w/ InfoNCE | $19.6 \pm 1.3$ | $0.851 \pm 0.027$ | $0.23 \pm 0.07$ | $0.14 \pm 0.02$ | 8.30 | 0.008 |
| Non-progressive w/ StableRep | $20.1 \pm 1.7$ | $0.852 \pm 0.024$ | $0.26 \pm 0.06$ | $0.14 \pm 0.02$ | 7.21 | 0.008 |
| Progressive - VicReg | $19.6 \pm 1.7$ | $0.852 \pm 0.036$ | $0.20 \pm 0.05$ | $0.13 \pm 0.02$ | 6.67 | 0.006 |
| Progressive - BYOL | $19.6 \pm 1.6$ | $0.856 \pm 0.032$ | $0.19 \pm 0.05$ | $0.13 \pm 0.02$ | 7.14 | 0.007 |
| Progressive w/ StableRep | $19.9 \pm 1.9$ | $0.850 \pm 0.028$ | $0.26 \pm 0.06$ | $0.14 \pm 0.02$ | 7.25 | 0.007 |

**Table 5:** Quantitative alignment results of the shared and video semantic spaces of the ablation experiments on SyncDraw-MM.

| Semantic space | Shared-space | | | | Video-space | | | |
|---|---|---|---|---|---|---|---|---|
| Model \ Metrics | CD↓ | HD↓ | $U_a↓$ | $U_b↓$ | CD↓ | HD↓ | $U_a↓$ | $U_b↓$ |
| Non-progressive | - | - | - | - | 2.42 | 1.565 | $-3.27$ | $-0.022$ |
| Non-progressive w/ InfoNCE | - | - | - | - | 3.16 | 1.406 | $-3.26$ | $-1.596$ |
| Non-progressive w/ StableRep | - | - | - | - | 3.13 | 1.425 | $-3.27$ | $-1.767$ |
| Progressive - VicReg | 0.04 | 0.220 | $-0.112$ | $-0.011$ | 2.06 | 1.328 | $-3.27$ | $-0.781$ |
| Progressive - BYOL | 0.02 | 0.153 | $-0.048$ | $-0.005$ | 2.06 | 1.318 | $-3.26$ | $-0.764$ |
| Progressive w/ StableRep | 1.13 | 1.219 | $-2.032$ | $-1.628$ | 3.13 | 1.428 | $-3.26$ | $-1.771$ |

## 5 Limitations

We explored a particular class of video architectures for representation learning based on autoencoder models. Other architectures such as the Vision Transformer (ViT) from VideoMAE (Wang et al., 2023) could be adapted with a full video decoder. Moreover, the representation learning method for the target modality could be extended to other approaches with different assumptions on the data distributions, such as relational regularization (Xu et al., 2020) and diffusion-based VampPriors (Kuzina & Tomczak, 2024), to understand their impact on the video semantic space. Additionally, different paradigms for addressing the contrastive multi-positive setup could be considered for adoption, such as probabilistic-based approaches (Chun et al., 2021) and auxiliary methods that support this modeling (Patrick et al., 2021).

We also focus on a scenario in which the video target modality is represented by fixed chunks of video. Future work can consider merging these chunks of video and preserving spatial and temporal consistency for a longer video generation based on a cross-modality approach with feature alignment. Furthermore, we focus our analysis on small to medium scale data sets for which the alignment is still not solved. Although the one-to-many case is also present in large data sets as shown in Section 4.2, future work could analyze this scenario to provide further insights into this problem subclass.

## 6 Discussion and Conclusions

We shed light on an implicit problem of cross-modality generation that is currently underexplored. Although modality alignment in a shared semantic space could benefit from knowledge obtained by pre-trained models for each modality, a challenging aspect arises related to how to map both modalities. For the one-to-many case, we propose a latent space analysis perspective for assessing alignment methods based on a progressive learning framework coupled with bucket loss to learn a shared semantic space between text and video modalities. We show that the one-to-many case presents varying levels of complexity across data sets, impacting the overall text-to-video generation results when using a shared semantic space. Moreover, this task lacks effective quantitative metrics, requiring complementary methods for robust assessment.

In this work, we focus on autoencoder models as they implicitly enable representation learning of the modality and can be adapted for cross-modality generation through feature alignment. We show how key components of this task affect the overall result and demonstrate that video representation plays an important role in

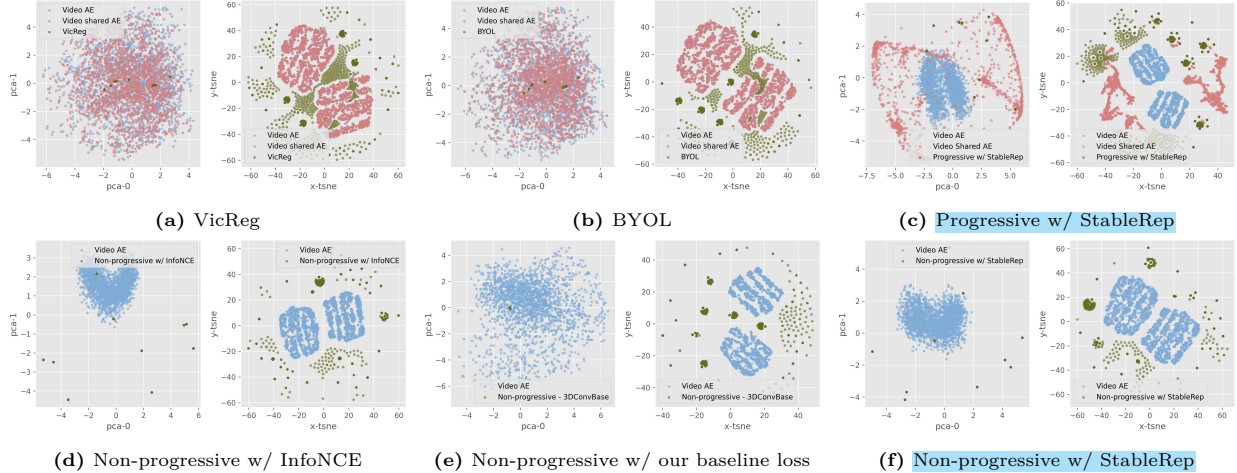

**Figure 8:** Video latent spaces from ablation experiments, showing embeddings from the mapping functions, video shared autoencoders, and video representation learning, visualized with PCA (odd columns) and t-SNE (even columns).

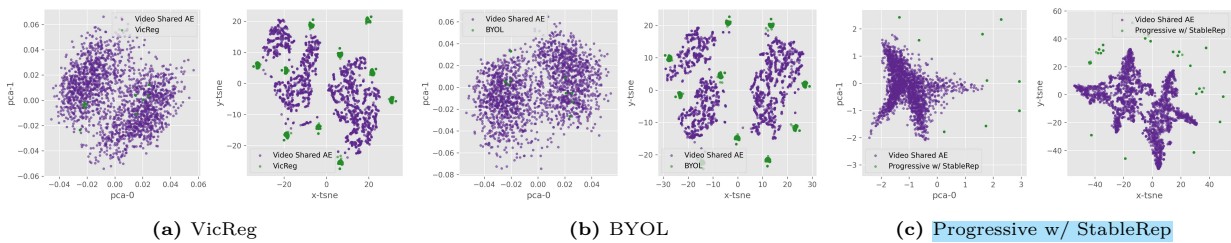

**Figure 9:** Shared semantic latent spaces from ablation experiments, showing embeddings from the mapping functions and video shared autoencoders, visualized with PCA (odd columns) and t-SNE (even columns).

the alignment process. Overall, tackling the one-to-many case is not straightforward, requiring a different perspective when considering a semantically shared space between modalities, as current methods and regularization techniques are not designed with this case in mind.

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

## Appendix

## A    Video Semantic Space

To generate videos, we need to learn a distribution $p(v|z_v)$ that is conditioned on our semantic space and that is similar to the original video data $p^*(v)$. Toward this goal, we minimize the Wasserstein distance between both distributions by using its dual form (Bousquet et al., 2017; Tolstikhin et al., 2018) of optimizing through random encoders $q(z_v \mid v)$ instead of the original distribution couplings. Hence, we minimize

$$D_{\mathrm{W}}\left(p^*(v), p(v \mid z_v)\right) = \inf_{q(z_v \mid v) \in \mathcal{Q}} \left\{ \mathbb{E}_{x \sim p^*(v)} \mathbb{E}_{z_v \sim q(z_v \mid v)} [c(x,y)] + \lambda_z \mathcal{D}(q(z_v), p(z_v)) \right\}, \quad (\text{A.1})$$

where $\mathcal{Q}$ is a non-parametric set of probabilistic encoders, $p(v \mid z_v)$ is our generative distribution, $x \sim p^*(v)$ is a ground truth video, $y \sim p(v \mid z_v)$ is a generated video (through decoder $G$) that depends on the semantic vector $z_v \sim q(z_v \mid v)$, and $\lambda_z > 0$ is a hyperparameter that weights the divergence measure $\mathcal{D}$ between the marginal distribution $q(z_v) = \mathbb{E}_{v \sim p^*(v)} [q(z_v \mid v)]$ and the prior $p(z_v)$ for our semantic space, and $c$ is a similarity cost.

**Video Similarity.** The cost function $c$ represents a measure between two videos, which we define as

$$c(x,y) = \lambda_{pixel}\big\|x - y\big\|_1 + \lambda_f \big\|f_{D_v}(x) - f_{D_v}(y)\big\|_1 + \lambda_p \big\|f_{\mathrm{VGG}}(x) - f_{\mathrm{VGG}}(y)\big\|_2^2, \quad (\text{A.2})$$

where $f_{D_v}(x)$ denotes the features of an intermediate layer of the video discriminator $D_v$, when considering video $x$; similarly, $f_{\mathrm{VGG}}(x)$ denotes the features of a VGG19 network (Johnson et al., 2016); and $\lambda_{pixel} > 0$, $\lambda_f > 0$, and $\lambda_p > 0$ are hyperparameters that define the weight of each term in the final cost.

This cost function penalizes the discrepancy between the videos on the pixel (left term) and feature space (middle to right term). The penalization on the feature space acts as a perceptual similarity measure between the original and generated samples, since pixel-wise metrics have difficulties capturing perceptual properties of the reconstructed samples. Our perceptual measure is defined as a feature-matching loss (Bao et al., 2017; Salimans et al., 2016) over feature space $f_{D_v}$ of discriminator $D_v$ and feature space $f_{\mathrm{VGG}}$. We introduce the details of $D_v$ later in this section.

**Video Latent Space Divergency.** The divergence $\mathcal{D}$ represents a cost on the difference between two given spaces. In the original WAE (Tolstikhin et al., 2018), this divergence is obtained using a GAN or Maximum Mean Discrepancy approach. In contrast, we consider a metric based on feature matching (Salimans et al., 2016), which we found to be more stable to train. We convert the WAE-GAN divergence (Tolstikhin et al., 2018), defined as a non-saturating loss (Fedus et al., 2018; Goodfellow et al., 2014), into a distance minimization problem between the semantic feature spaces, $f_{D_z}$, of both $q(z_v)$ and $p(z_v)$. We empirically found that removing the min-max between the autoencoder (i.e., $E_v$ and $G$) and the discriminator $D_z$ led to a more stable training compared to the original WAE-GAN loss. Adding a gradient penalty (Fedus et al., 2018; Gulrajani et al., 2017) also leads to stable training, but we found that the feature matching term was enough to stabilize video training. Hence, we define the divergence as the aggregate

$$\mathcal{D}(q(z_v), p(z_v)) = \mathcal{L}_f + \mathcal{L}_{D_z} + \mathcal{L}_{D_v}, \quad (\text{A.3})$$

where the losses $\mathcal{L}_{(\cdot)}$ depend on the same arguments as $\mathcal{D}$. The feature-matching loss $\mathcal{L}_f$ penalizes the semantic feature space induced by discriminator $D_z$, when it learned to distinguish between the true and a variational approximation of the semantic distributions. The video adversarial loss, $\mathcal{L}_{D_v}$, measures the similarity in the perceptual space as similar videos will have similar underlying semantic distributions, and the semantic discriminator loss, $\mathcal{L}_{D_z}$, induces similarity between prior and approximated semantic distributions.

We consider the feature-matching loss as

$$\mathcal{L}_f(q(z_v), p(z_v)) = \mathbb{E}_{\tilde{z}_v \sim p(z_v)} \mathbb{E}_{z_v \sim q(z_v)} \big\|f_{D_z}(\tilde{z}_v) - f_{D_z}(z_v)\big\|_2^2, \quad (\text{A.4})$$

such as $f_{D_z}(z_v)$ denotes the features of an intermediate layer of $D_z$ when considering the latent vector $z_v$, and the joint semantic space $p(z_v)$ is modeled as a multivariate normal distribution.

**Table B.1:** Data set splits used for training and testing containing the number of text and video pairs along with its corresponding number of buckets.

| Data Set | Train | Validation | Test | Buckets |
|---|---|---|---|---|
| SyncDraw-MM | 10000 | 2000 | 2000 | 20 |
| KTH | 21030 | 5502 | 6650 | 150 |
| TACoS | 31392 | 7848 | 9811 | 11659 |
| Panda70M | - | - | 101599 | 5178 |

Then, we define the semantic discriminator loss to penalize the difference between the true distribution, $p(z_v)$, and our approximation, $q(z_v)$, as

$$\mathcal{L}_{D_z} = - \mathbb{E}_{\tilde{z}_v \sim p(z_v)} \left[ \log D_z(\tilde{z}_v) \right] - \mathbb{E}_{z_v \sim q(z_v)} \left[ \log(1 - D_z(z_v)) \right], \tag{A.5}$$

where $D_z$ is the semantic space discriminator. Finally, the video discriminator $D_v$, from which we compute $f_{D_v}$ in Equation A.2, tries to differentiate between real, $p^*(v)$, and generated videos, $p(v \mid z_v)$, with a loss similar to Equation A.5 but now considering the videos samples instead of the semantic vectors.

## B    Data Sets

Moving MNIST is an extension of the MNIST (Lecun et al., 1998) data set where one or two digits move up and down, left to right, and vice versa. Each video has a sentence describing the digits and their moving direction. For the KTH data set, which contains videos of several actions of 25 persons recorded in four different backgrounds with variations in light and clothing, we selected a subset of these actions (i.e., walking, jogging, and running) as in Mittal et al. (2017) and Marwah et al. (2017) experiments. We also provide a new set of text descriptions for this data[3]. Each text description indicates the person in the video, its corresponding action, and direction of movement, such as "person 2 is walking left to right" and "person 5 is jogging right to left." The last category-oriented set, the TACoS data set contains videos of people cooking with multilevel descriptions, such as one sentence, short, and detailed descriptions for each video. In our experiments, we selected the set of short descriptions that better represents the *one-to-many* case, with each description depicting an event over a time interval in the video.

Moreover, training and test buckets are disjoint, i.e., the same bucket does not appear in both sets, though buckets may share attributes such as movement direction. For example, in SyncDraw-MM, the bucket "the digit 2 is moving left and right" appears only in training, while the same movement pattern with unseen digits (e.g., "the digit 4 is moving left and right") appears only in testing. Similarly, in KTH, person identities in the test split do not appear in training. For example, "person 16 is running left to right" is available in the test set, but "person 16" is absent from training, though other persons performing the same movement are present.

For Panda70M (Chen et al., 2024), which is built on a selection of videos from HD-VILA-100M (Xue et al., 2022), we considered its test set that originally presents 2k long-length videos. From this list, only 1843 videos were available for download. We selected only the video periods containing associated text description that generated 5097 valid video segments, for which the video splitting generated 101599 video chunks. We present data sets splits in Table B.1 and bucket examples in Figure B.1.

## C    Implementation Details

We present additional implementation details used for latent space analysis, metrics, and model architectures.

---

[3]Mittal et al. (2017) and Marwah et al. (2017) also generated a set of text descriptions for the KTH data set, but they are not publicly available.

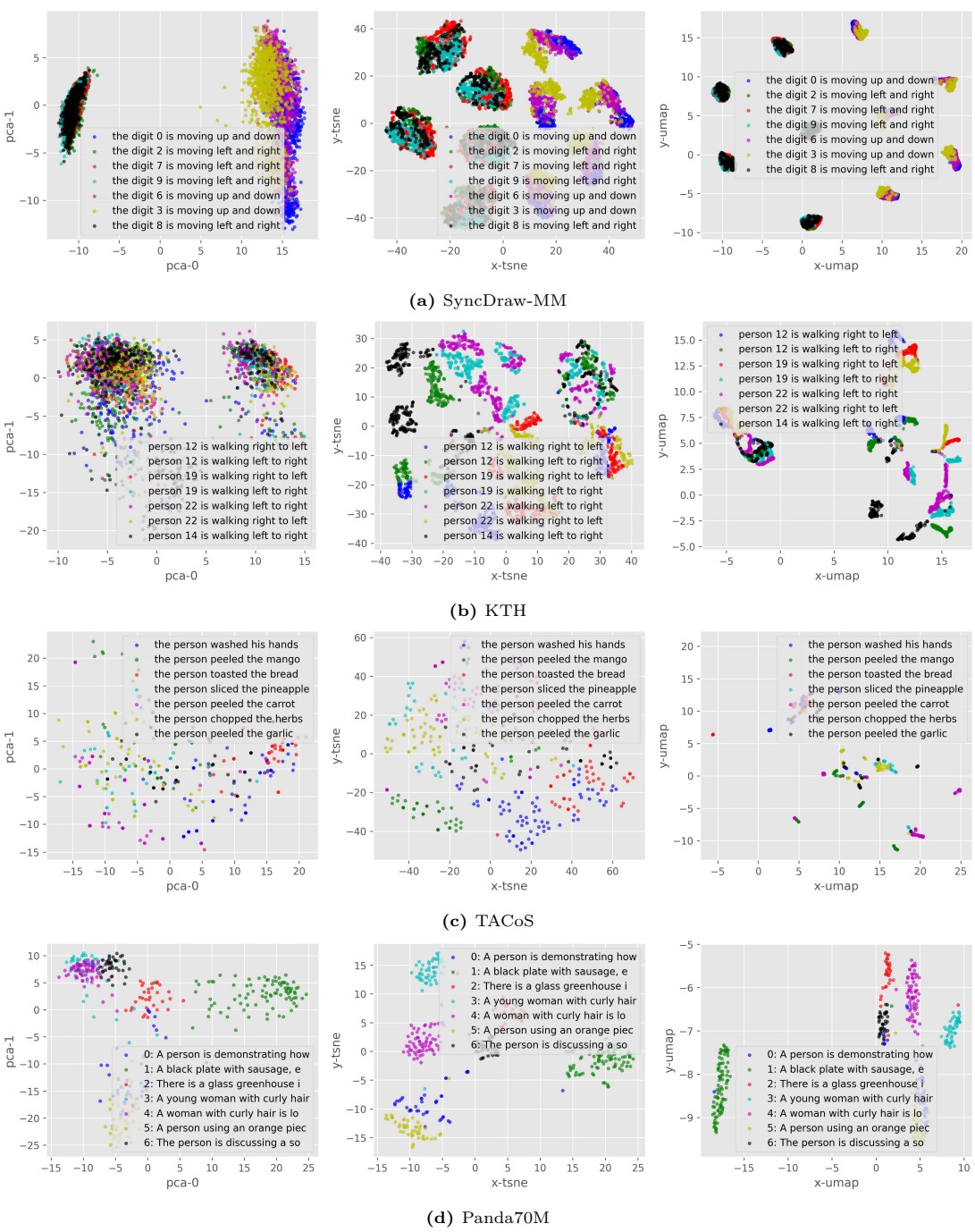

**Figure B.1:** Samples of seven random buckets visualized in the video latent spaces generated using VideoMAE-v2 (Wang et al., 2023) encoder, for each data set. The columns results correspond to: PCA, t-SNE, and UMAP, respectively. The buckets presented for Panda70M have the phrases trimmed for illustration purposes and contain the following: 'A person is demonstrating how to put a card into a usb micro usb slot on a device'; 'A black plate with sausage, eggs, and cheese'; 'There is a glass greenhouse in the backyard with plants inside'; 'A young woman with curly hair is walking down the street at night'; 'A woman with curly hair is looking at the camera'; 'A person using an orange piece of tape to hold a marker'; and 'The person is discussing a soccer player's training and health'.

### C.1   Latent Space Analysis

The t-SNE visualization is produced by first reducing the input dimensionality to 32 components with PCA, and then applying t-SNE over the resulting components with a perplexity of 40 and a number of iterations equal to 600 for all visualizations. Both t-SNE (van der Maaten & Hinton, 2008) and UMAP (McInnes et al., 2018) are stochastic methods with the goal of preserving local structure. In order to reproduce their initial randomness process, a random state variable can be used in these methods (e.g. seed of 42).

For the VideoMAE (Tong et al., 2022; Wang et al., 2023) representation used for the analysis of the overall structure of the one-to-many case, we selected the VideoMAE-v2 (Wang et al., 2023) model, more specifically the Hybrid-PT-SSv2-FT version used in Ge et al. (2024)[4]. The ViT-g encoder features were extracted following their guidelines, generating embeddings with dimension 1408 from the penultimate layer of the encoder that were averaged across all patches.

### C.2   Metrics

We used the official implementation of LPIPS (Zhang et al., 2018)[5] and the AlexNet (Krizhevsky et al., 2012) backbone to calculate the metric. Other parameters were defined with the default values used in the official code. We used the official implementation of DISTS (Ding et al., 2022)[6] and the PyTorch version of the metric. The default backbone used was based on VGG16 (Simonyan & Zisserman, 2015) with the default repository parameters.

For distribution-based metrics, we considered the following: FVD is calculated with the I3D video features (Carreira & Zisserman, 2017) extracted from the model (RGB stream) available on Kinects-I3D[7] with an extension of the FID metric from Heusel et al. (2017)[8]; and KVD with the polynomial MMD (Unterthiner et al., 2018).

For the alignment metrics, we used the PyTorch3D implementation for the Chamfer distance[9] and the Scipy based implementation for Hausdorff[10] which is used to calculate a symmetric version. The features are L2-normalized before the distance calculation. The uniformity is adapted from the official implementation[11] where points are L2-normalized onto the unit hypersphere first and considering a maximum of 5000 points randomly selected.

### C.3   Architectures

We present details of each training setup for text, video and the progressive decoupling method. Additionally, Table C.1 shows an overview of the number of parameters of the models used for the video autoencoder, mapping function and video semantic shared autoencoder.

#### C.3.1   Text Models

We evaluated the CLIP (Radford et al., 2021) text encoder, which is used with its pre-trained model from the ViT-B/32 version. The CLIP method used was based on the `transformers` package[12] using the pre-trained model with key `openai/clip-vit-base-patch32` generating a 512-dimensional embedding.

The word dictionary used as the noise set for sampling a noise word for the text in the cross-modal alignment was built based on DBPedia (Lehmann et al., 2015)[13] and is processed similarly to Dai & Le (2015). First,

---

[4]https://github.com/songweige/content-debiased-fvd
[5]https://github.com/richzhang/PerceptualSimilarity
[6]https://github.com/dingkeyan93/DISTS
[7]https://github.com/google-deepmind/kinetics-i3d
[8]https://github.com/bioinf-jku/TTUR/
[9]https://pytorch3d.readthedocs.io/en/latest/modules/loss.html#pytorch3d.loss.chamfer_distance
[10]https://docs.scipy.org/doc/scipy/reference/generated/scipy.spatial.distance.directed_hausdorff.html
[11]https://github.com/ssnl/align_uniform
[12]https://huggingface.co/docs/transformers/en/model_doc/clip#transformers.TFCLIPTextModel
[13]Downloaded from https://github.com/srhrshr/torchDatasets/. The data set splits ('train' and 'test') provided were the ones used in our experiments as well.

we treat punctuation as separate tokens. Then, we ignore any non-English characters and words. Since the removal of non-English words can affect the semantics of the text, we also remove entries that have too many `UNKNOWN` tokens after this preprocessing. We have defined a maximum value of 45% of unknown tokens to be considered a valid entry for the set. We also remove words that appear only once in the set, and we do not perform any term weighting or stemming in the preprocessing. This word dictionary with the exception of words in each data set is the final dictionary set used.

### C.3.2 Video Models

For the video pixel-based discriminator $D_v$, we adapted the Patch discriminator from Pix2PixHD (Wang et al., 2018) that evaluates video quality on multiple scales. For the video representation, we considered the dimensions: $d_z = 64$ for 3DConv-Base and $d_z = 128$ for UNetLDM. Other regularization coefficients were defined as $\lambda_{pixel} = 10$, $\lambda_f = 10$, $\lambda_z = 5$. In particular, we defined $\lambda_p = 0.0025$ since this term dominated other terms in the final loss and this value presented satisfactory results in perceptual quality. In this case, the perceptual weight is defined over the VGG19 layers: `block4_conv3` and `block5_conv4`. The training setup for the video autoencoder considered Adam optimizer with a learning rate of $10^{-4}$ with a global clip norm (maximum gradient norm of 5.0). We trained the video models for about 100 epochs with varying batch size of $32 - 100$, for UNetLDM and 3DConvBase, respectively.

For the video cost in Equation A.2, we found empirically that an L1-based distance converged better for the pixel and feature discriminator terms, while an L2-square distance worked better for perceptual loss.

### C.3.3 Progressive Decoupling

In the decoupling process, we also consider a second text description input $\hat{t}_i$ from $t_i$, where a noise word is added with probability $p = 0.15$ to include variation in text representation in the same bucket $b_i$, but having the bucket loss considering the original text embeddings $t_i$. The word dictionary from which the noise sample is obtained did not include any words from the corresponding data set corpus. We also evaluated dropout noise (Gao et al., 2021), but empirically found that the addition of random words worked better. Word removal, on the other hand, was not suitable as it directly interferes with the original bucket semantics since removing some words could join samples from originally different buckets.

For the progressive decoupling architecture, we considered a multilayer perceptron (MLP) with four layers. Except for the last layer, each was defined with a hidden layer size of 512 and is followed by a Layer Normalization (Ba et al., 2016) and Swish activation function (Ramachandran et al., 2017). A dropout layer is used after the second and third layers with a rate of 0.1. This was the base network used for the mapping function and video shared autoencoder, changing only the input and output dimensions to match the corresponding representation sizes. The regularization coefficients were defined as $\lambda_{z_s} = 5.0$, $\lambda_s = 100$, $\lambda_{feat} = 10$, $\lambda_{pixel}^s = 30$. In addition, for the bucket loss, we define $\lambda_{neg} = 1.0$, $\lambda_{pos} = 1.0$, $\alpha = 2.0$ to weight the direct text-video pairs of the bucket, and the threshold over the pairwise cosine similarities to consider a sample belonging to the same bucket or not as 0.999. The training setup also considered Adam optimizer with a learning rate of $10^{-4}$ with a global clip norm (maximum gradient norm of 4.0). We trained the models for about 70 epochs with varying batch size of $32 - 100$, which depends on the video autoencoder used and the cross-modality alignment approach.

## D  Additional Results

We present additional results on the video autoencoder models and the progressive decoupling ablation experiments.

### D.1  Video Representation Learning

In Figure D.1, we present the latent spaces of the 3DConv-Base and UNetLDM video autoencoder models separately, whereas in Section 4.3 we presented their joint latent space. Figure D.2 also shows two different

**Table C.1:** Size of the networks and components used in this work.

| Model | Number of parameters |
|---|---|
| 3DConv-Base | 9.9M |
| UNetLDM | 268.8M |
| VDM (Ho et al., 2022) | 35.7M |
| Mapping function $M$ ($d_{z_t} = 512$ and $d_{z_s} = 64$) | 824k |
| Video semantic shared AE ($E_s$ and $G_s$) ($d_{z_v} = 64$ and $d_{z_s} = 64$) | 1.2M |
| Components | |
| Discriminator $D_s$ or $D_z$ with $d_z = 64$ | 133K |
| Discriminator $D_v$ (PatchHD-Video) | 2.6M |
| VGG19 (Johnson et al., 2016) network | 20M |

runs of PCA, t-SNE and UMAP visualization for the same latent space that exhibits the different overall distribution structures obtained in these different initializations.

In Figure D.3, we present the qualitative results for the video autoencoders: 3DConv-Base, UNetLDM, and VDM (Ho et al., 2022). From the SyncDraw-MM set, we observed better quality with UNetLDM. The 3DConv-Base model generates correct results, but has more misleading cases and lacks sharpness in some cases. In the SyncDraw-MM set, for example, there are cases where digit 5 is misplaced with 3, or 9 with 4, and 1 with 7. This occurs at a lower level in the other models. The VDM model, on the other hand, is not consistent with its results, with its major drawback being the lack of filling in the digit (e.g., holes in some digits) and the thin look in most samples. This model also does not correctly generate the digits in a large part of the samples, generating instead frames with black background and random white points in the border without any digit enclosed.

For the KTH set, UNetLDM also produces sharper videos compared to 3DConv-Base, which in some cases generates an artifact resembling an aura over the person. In this set, UNetLDM appears to produce a brighter background as well. VDM model produces sharper videos and also a large diversity in the samples, but following the results with the previous set, there is a large amount of poor samples generated where there is no movement or person in the video.

Lastly, for the TACoS set, 3DConv-Base generated videos with less fine-grained details. For some cases, this seems to impact the understanding of the movement depicted in the video. The aura effect also occurs in some samples of this set around people. The UNetLDM model generated more blur effects for TACoS and some artifacts resembling "checkboard" artifacts, mostly in brighter parts. The VDM model produced sharper videos for this set, and it was observed that the majority of the samples were generated with people in darker clothes.

### D.1.1 Ablation Experiments

We performed ablation experiments on video representation learning, where the quantitative results are presented in Table D.1, qualitative results are shown in Figure D.4, and latent spaces are shown in Figure D.5. We evaluated three main components using the 3DConv-Base architecture: the impact of the video latent dimension size; the impact of the autoencoder type by comparing with a Variational Auto-Encoder (VAE); and the impact of the distribution discriminator $D_z$ in the WAE-GAN-based approach.

From the quantitative results, we observe that the dimension size variation ($d_z$) does not significantly affect the quantitative results. However, switching from the WAE-GAN-based approach to VAE degrades reconstruction. Removing the distribution discriminator $D_z$ also degrades performance, except for the VAE approach, which achieves better results without it.

The qualitative results reveal inferior reconstruction with the plain VAE approach where more than one digit appears to be reconstructed, resulting in a mirror effect. Additionally, most videos seem to be concentrated on "up and down" movements rather than "left to right" movements. However, this effect appears partially associated with the distribution discriminator $D_z$, as removing it improves qualitative results on the same

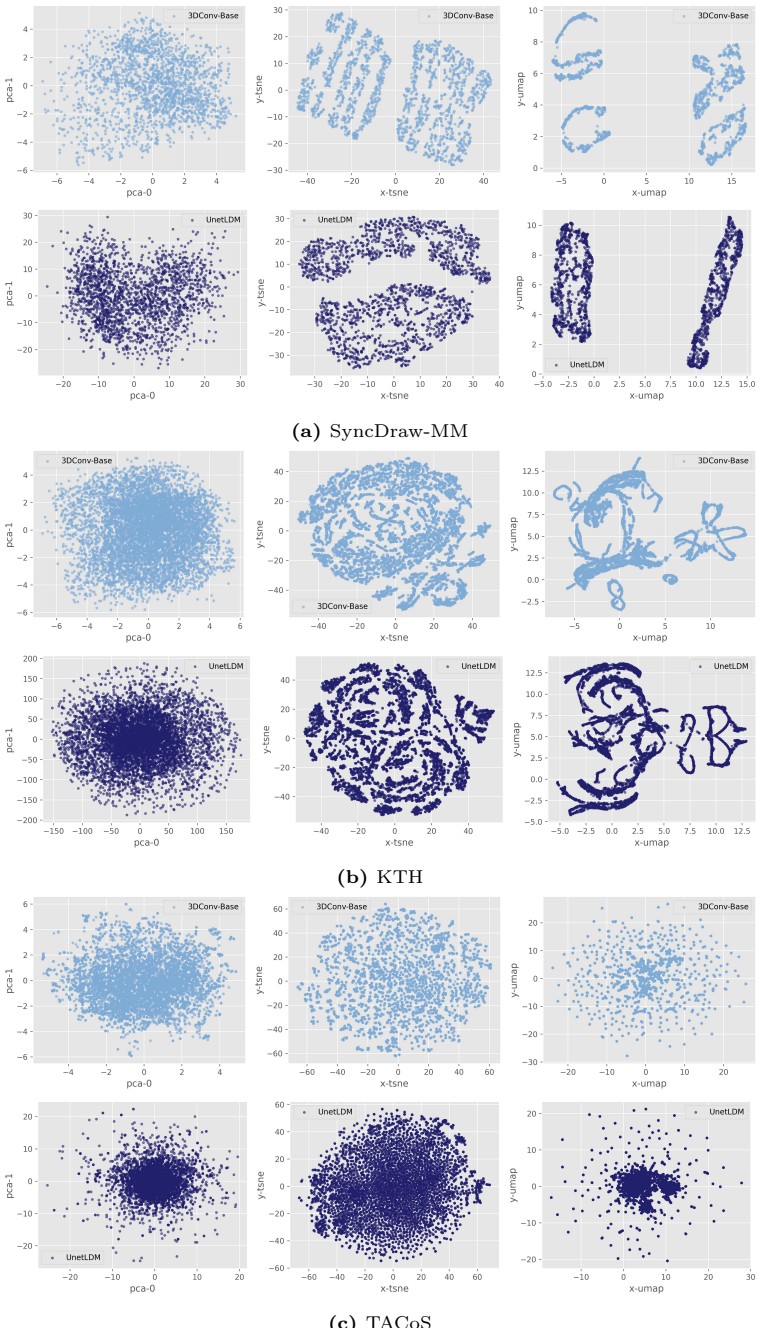

**Figure D.1:** Video semantic spaces obtained with the 3DConv-Base (first rows) and UNetLDM (second rows) models for the Syncdraw-MM, KTH, and TACoS data sets.

instances. When removing $D_z$ from our main approach, we observe a small decrease in some full-reference metrics (e.g., PSNR) but slightly better distribution-based metrics (e.g., FVD). Qualitatively, this difference is minimal, as the videos from both approaches are similar, with both showing some loss of fine-grained details in the digits. Regarding latent dimension size, we observe minor differences in fine-grained details between models on the same instances.

Moreover, examining the video latent spaces in Figure D.5, we observe slight structural differences across models. For dimension variation, latent spaces with $d_z \geq 128$ are sparser than those with $d_z < 128$ (also

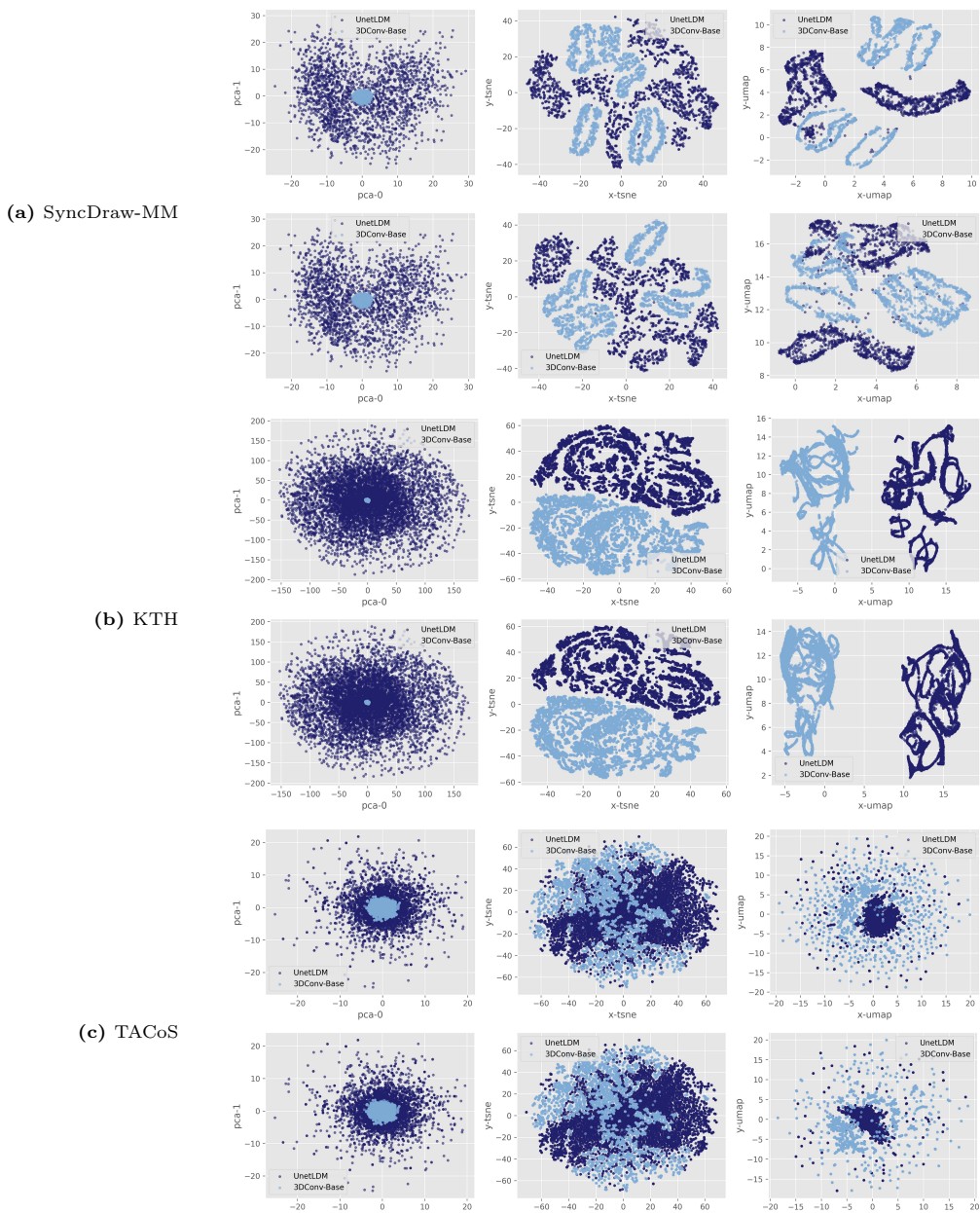

**(a)** SyncDraw-MM

**(b)** KTH

**(c)** TACoS

**Figure D.2:** Two runs with different initialization seeds (first and second rows) for the visualization of the video semantic spaces from 3DConv-Base and UNetLDM on the SyncDraw-MM, KTH, and TACoS sets. For UNetLDM, we apply PCA to reduce dimensionality to match the dimension of 3DConv-Base. Initialization seeds are considered for t-SNE and UMAP only.

including $d_z = 64$ in Figure D.1). In contrast, VAE-based approaches produce more concentrated latent spaces when we evaluate their distribution with PCA. Additionally, removing the distribution discriminator $D_z$ alters the latent space structure compared to the baseline in Figure D.1.

## D.2 Progressive Decoupling Learning

In Figures D.6, D.7, and D.8, we present qualitative results for text-to-video generation produced with the alignment models of Section 4.4 for: SyncDraw-MM, KTH, and TACoS data sets.

**Table D.1:** Quantitative results of the video ablation experiments performed with SyncDraw-MM and the 3DConv-Base video architecture evaluating the impact of: dimension size of latent space $d_z$ and the general autoencoder adopted approach.

| Metrics | PSNR↑ | SSIM↑ | LPIPS↓ | DISTS↓ | FVD↓ | KVD↓ |
|---|---|---|---|---|---|---|
| | | | Dimension | | | |
| $d_z = 48$ | $19.2 \pm 1.9$ | $0.89 \pm 0.03$ | $0.08 \pm 0.02$ | $0.09 \pm 0.02$ | 2.81 | 0.003 |
| $d_z = 128$ | $19.0 \pm 1.8$ | $0.887 \pm 0.030$ | $0.09 \pm 0.02$ | $0.09 \pm 0.02$ | 2.77 | 0.003 |
| $d_z = 256$ | $19.1 \pm 1.9$ | $0.889 \pm 0.030$ | $0.08 \pm 0.02$ | $0.09 \pm 0.02$ | 2.75 | 0.003 |
| | | | General autoencoder approach | | | |
| Base w/o $D_z$ | $18.8 \pm 1.8$ | $0.885 \pm 0.030$ | $0.09 \pm 0.02$ | $0.09 \pm 0.02$ | 2.60 | 0.003 |
| VAE w/o $D_z$ | $17.3 \pm 1.2$ | $0.860 \pm 0.020$ | $0.11 \pm 0.04$ | $0.11 \pm 0.02$ | 3.34 | 0.004 |
| VAE | $15.3 \pm 1.2$ | $0.757 \pm 0.040$ | $0.29 \pm 0.07$ | $0.16 \pm 0.02$ | 9.10 | 0.017 |

From the SyncDraw-MM results, we observe a more difficult alignment task between the sets. All models exhibit poor video generation with a lack of fine-grained details for the digits. The worst results are those of the CoDi-based and our method with UNetLDM, both showing stronger indicators of representational collapse: the former generates only a sparse set of points representing the digits without proper content generation, while the latter produces nearly identical videos regardless of the input text. Overall, better results were observed for vertical movements compared to horizontal ones, although both are equally represented in the data set.

For the KTH set, some models present better results, which is possibly related to the lower one-to-many difficulty level of this set. We still observe a level of representational collapse, but to a lesser extent than with the SyncDraw-MM set. For CLIP-based and our method with 3DConv-Base, we observe an aura effect in some subjects, also observed in the video autoencoder results, suggesting this effect propagates from the video autoencoder. In contrast, our method with UNetLDM exhibits fewer of these artifacts. However, CoDi-based and ImageBind-based alignment present poor results, with blurrier frames that fail to depict the person performing the action described in the input text.

For the TACoS set, although the generated videos better depict the subjects than in the KTH set — strongly indicating a correlation with the difficulty level of the alignment — the results for some models exhibit representational collapse. The only exceptions are our method with 3DConv-Base video autoencoder and CLIP-based alignment. Our baseline with UNetLDM, previously found to have a representational collapse problem in latent space, also shows this effect in the generated videos. CoDi-based results also show a level of collapse, whereas ImageBind seems to better represent subjects, although it presents poor results with missing subjects, along with CoDi and our baseline with UNetLDM. An aura effect is also observed for models based on 3DConv-Base video autoencoder. The best-performing models are the CLIP-based alignment and our baseline with 3DConv-Base, though blur and the aura effect are more pronounced in the CLIP-based method.

### D.2.1 Ablation Experiments

In Table D.2, we present the video quantitative results of additional ablation on the cross-modality alignment with the best architecture found in the video representation ablation, and in Table D.3 their corresponding alignment quantitative results. Figure D.9 shows the corresponding latent spaces of the video and shared semantic representations, while in Figure D.10 we present the corresponding qualitative results.

This ablation shows the impact of representation learning on the mapping between modalities. We note that the structures of latent spaces change when we change the way the target modality is represented. Although it is primarily outstanding in shared semantic spaces, the structure is affected by producing sparser spaces (e.g., $d_z = 256$). Comparing the alignment quantitative results with the baseline model of Table 3, we observe that uniformity across both shared and video spaces is similar in both experiments. The overall video space alignment remains essentially unchanged, suggesting that $d_z = 256$ does not appear to provide a clear benefit in this scenario. However, $d_z = 64$ appears to yield a better aligned shared space.

**Table D.2:** Quantitative results of additional ablation experiments on the feature alignment between text and video modalities on the SyncDraw-MM data set with 3DConv-Base video autoencoder.

| Model/Metrics | B-PSNR↑ | B-SSIM↑ | B-LPIPS↓ | B-DISTS↓ | FVD↓ | KVD↓ |
|---|---|---|---|---|---|---|
| Dimension $d_z = 256$ | $19.7 \pm 1.9$ | $0.864 \pm 0.027$ | $0.27 \pm 0.10$ | $0.14 \pm 0.02$ | 6.98 | 0.006 |

**Table D.3:** Quantitative alignment results of the shared and video semantic spaces created for the additional ablation experiments on SyncDraw-MM set.

| Semantic space | Shared-space | | | | Video-space | | | |
|---|---|---|---|---|---|---|---|---|
| Model \ Metrics | CD↓ | HD↓ | $U_a$↓ | $U_b$↓ | CD↓ | HD↓ | $U_a$↓ | $U_b$↓ |
| Dimension $d_z = 256$ | 0.036 | 1.177 | $-0.791$ | $-0.696$ | 2.138 | 1.312 | $-3.25$ | $-0.682$ |

In the qualitative results, we observed poor generation, indicating a poor alignment level for the SyncDraw-MM set. The non-progressive method trained with our adapted loss shows a higher indicator of representational collapse, since their generated videos seem to follow, with small differences, the same outlined video. The non-progressive version with InfoNCE and with StableRep seems to suffer less with the representation collapse, although the video quality still lacks fine-grained details of the digits.

The VicReg seems to work better with particular digits, such as digit 4. This is similar to the BYOL results in this regard. For the video autoencoder 3DConv-Base with $d_z = 256$, we observe finer details for some digits, though still far from correct alignment. In general, one case noticed in the results was that the models seem to correctly follow the target motion, being better at the "moving up and down" category than the "left to right" or vice versa. Considering that SyncDraw-MM set is balanced in this regard, i.e., the number of videos with the "moving up and down" is close to the number of videos of "moving left to right" (or vice versa), this can show a more difficult alignment in the later large bucket of horizontal movement.

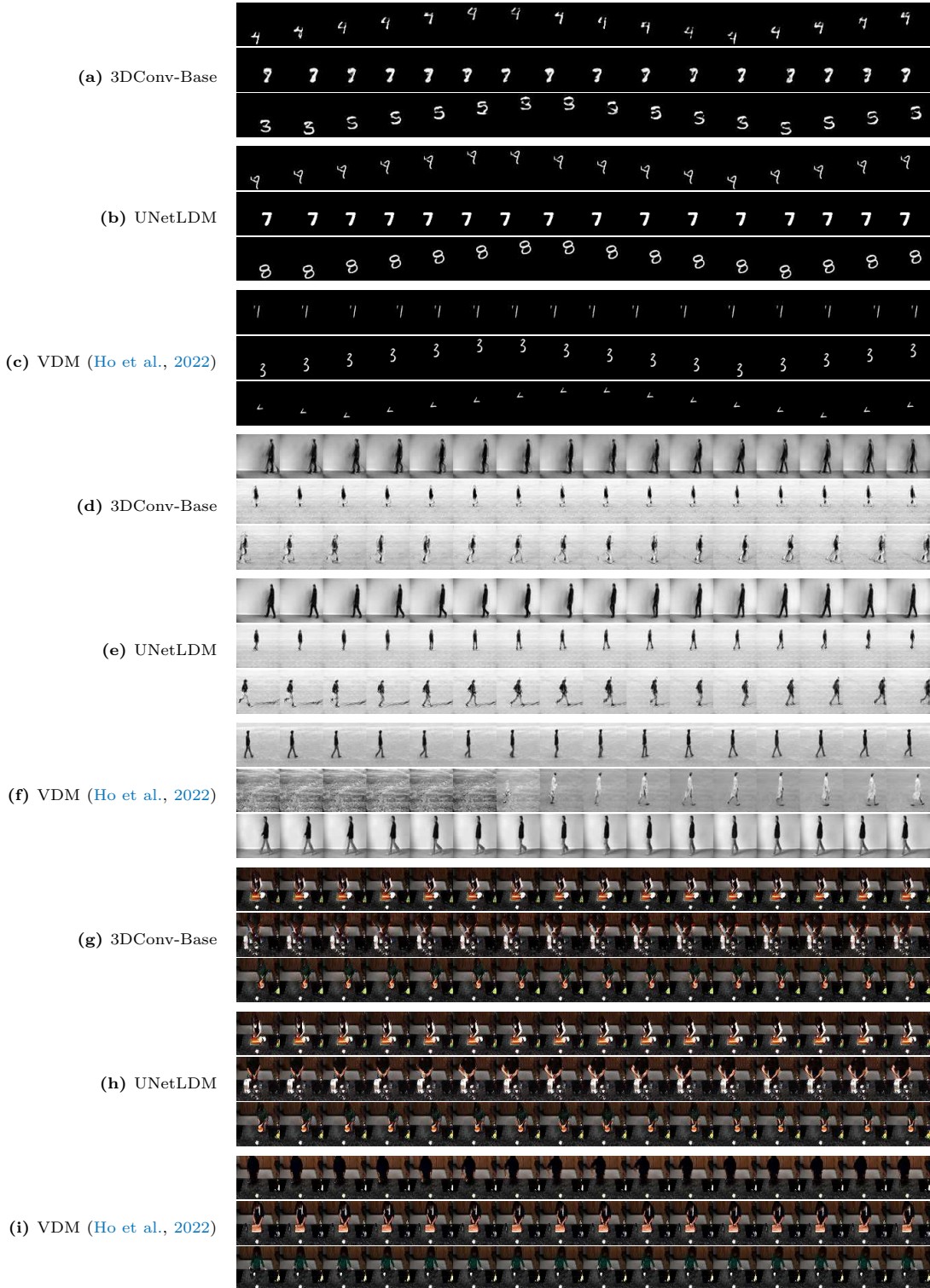

**Figure D.3:** Comparison of the generated videos by the video autoencoder models on the SyncDraw-MM (a-c), KTH (d-f), and TACoS (g-i) data sets with 3DConv-Base, UNetLDM, and VDM (Ho et al., 2022).

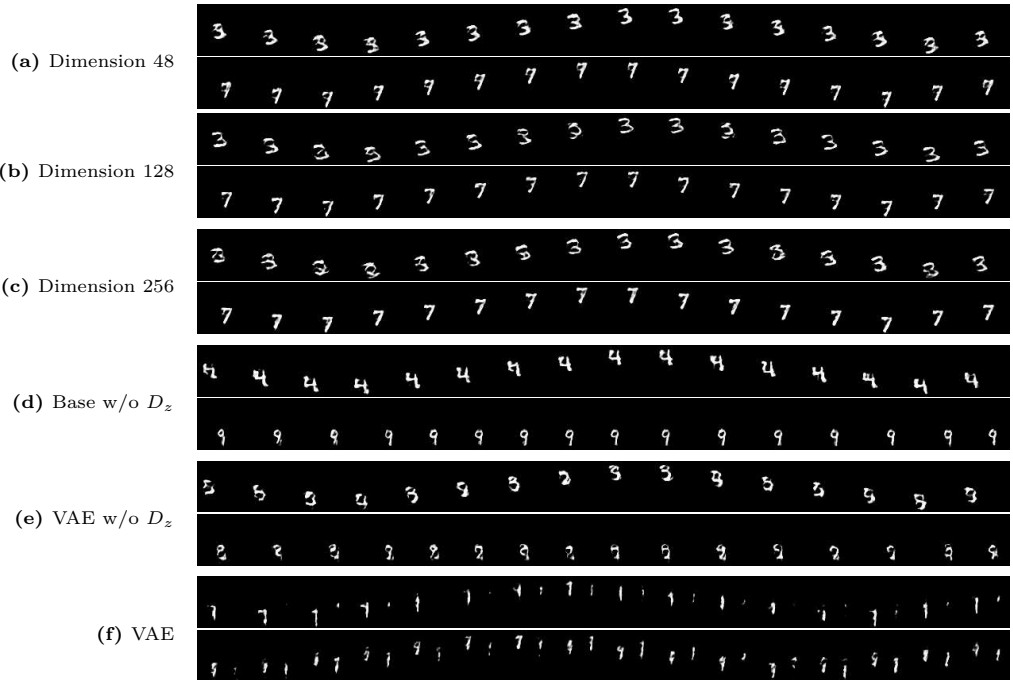

**Figure D.4:** Comparison of the generated videos by the video autoencoder models from the ablation experiments on the SyncDraw-MM set with 3DConv-Base architecture: variations on latent space dimension and autoencoder approach.

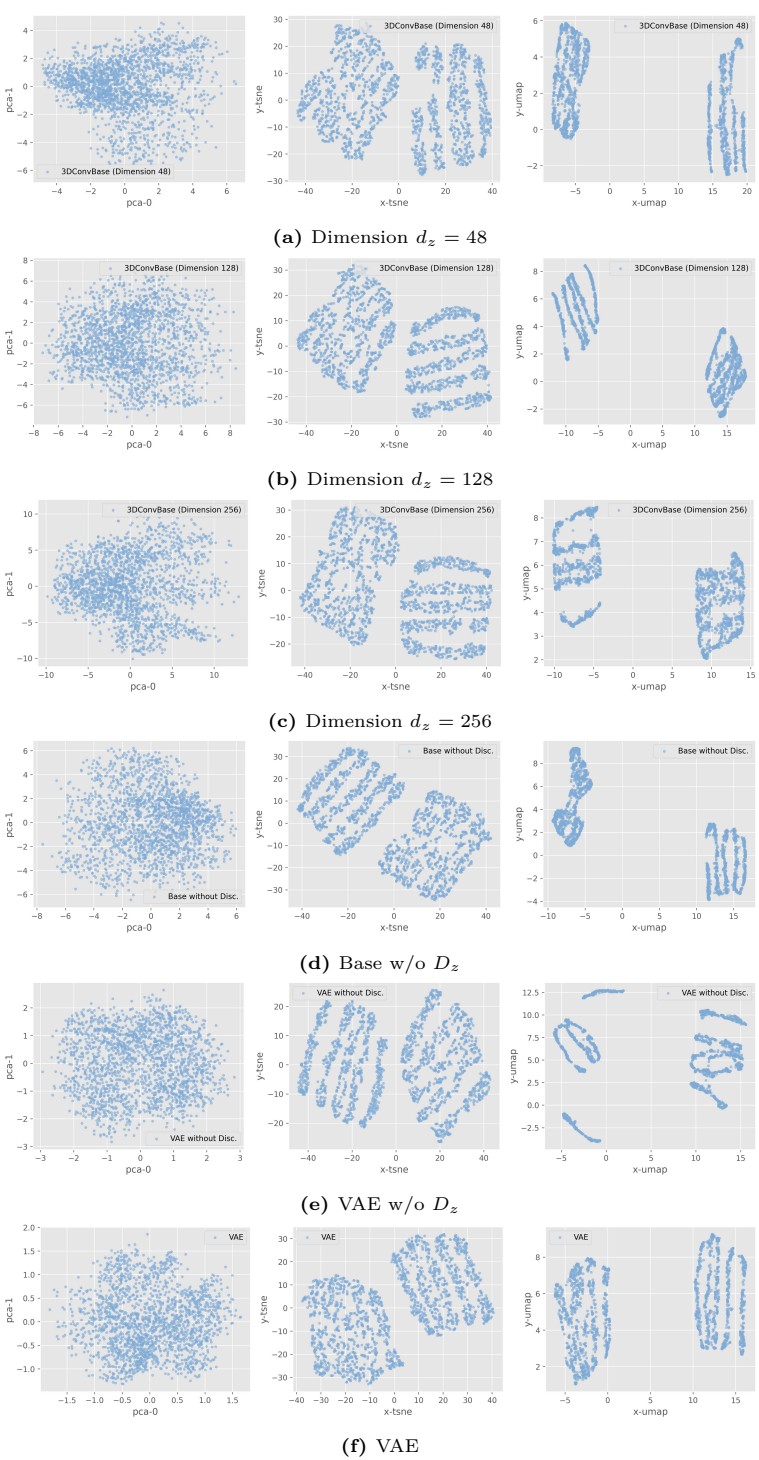

**(a)** Dimension $d_z = 48$

**(b)** Dimension $d_z = 128$

**(c)** Dimension $d_z = 256$

**(d)** Base w/o $D_z$

**(e)** VAE w/o $D_z$

**(f)** VAE

**Figure D.5:** Video semantic spaces obtained using 3DConv-Base for video ablation experiments on: (a-c) latent dimension $d_z$, (d) removal of distribution discriminator $D_z$, and (e-f) VAE approach.

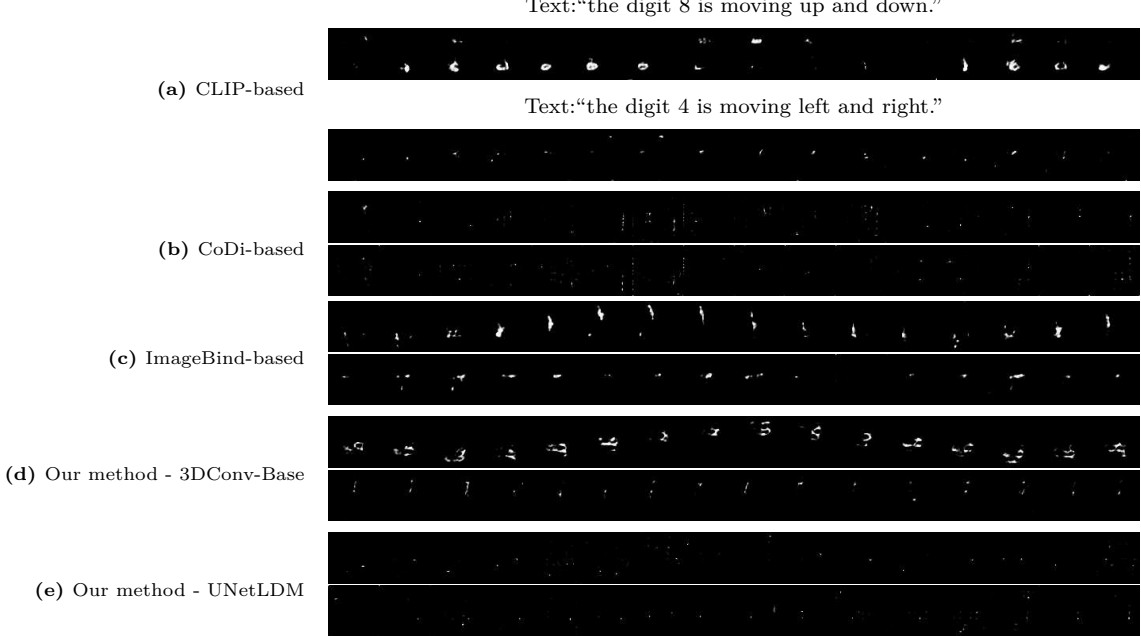

**Figure D.6:** Comparison of the generated videos by the alignment models: CLIP-based, CoDi-based, ImageBind-based, our baseline method with both 3DConv-Base and UNetLDM video architectures, on the SyncDraw-MM data set.

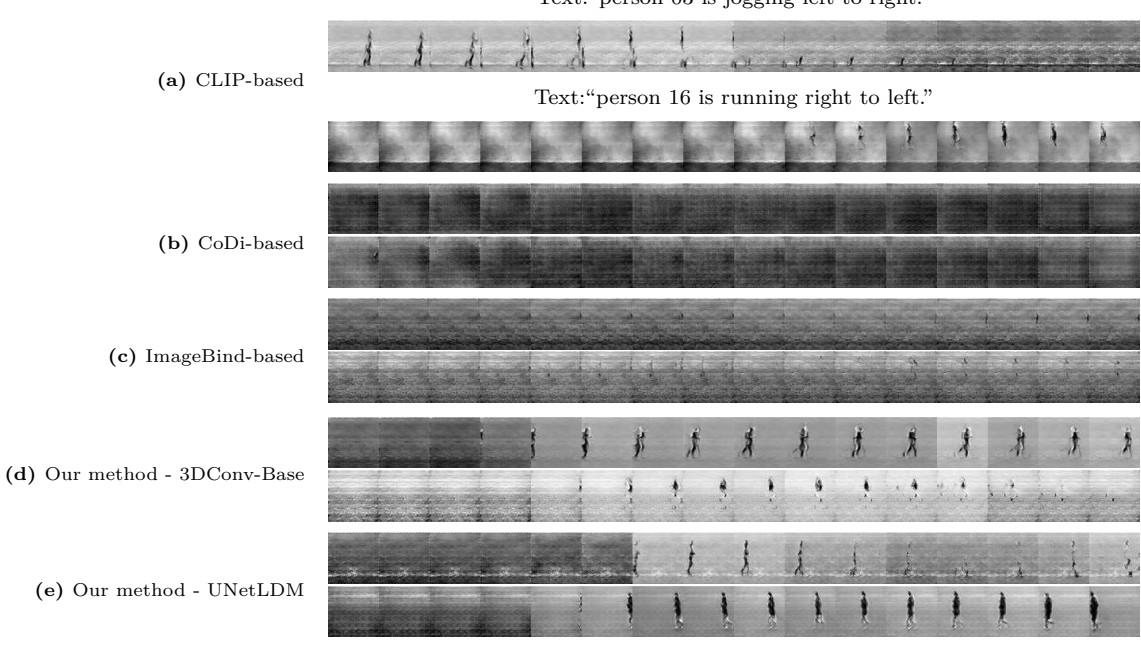

**Figure D.7:** Comparison of the generated videos by the alignment models: CLIP-based, CoDi-based, ImageBind-based, our baseline method with both 3DConv-Base and UNetLDM video architectures, on the KTH data set.

Text:"the person cut those beans into very small pieces."

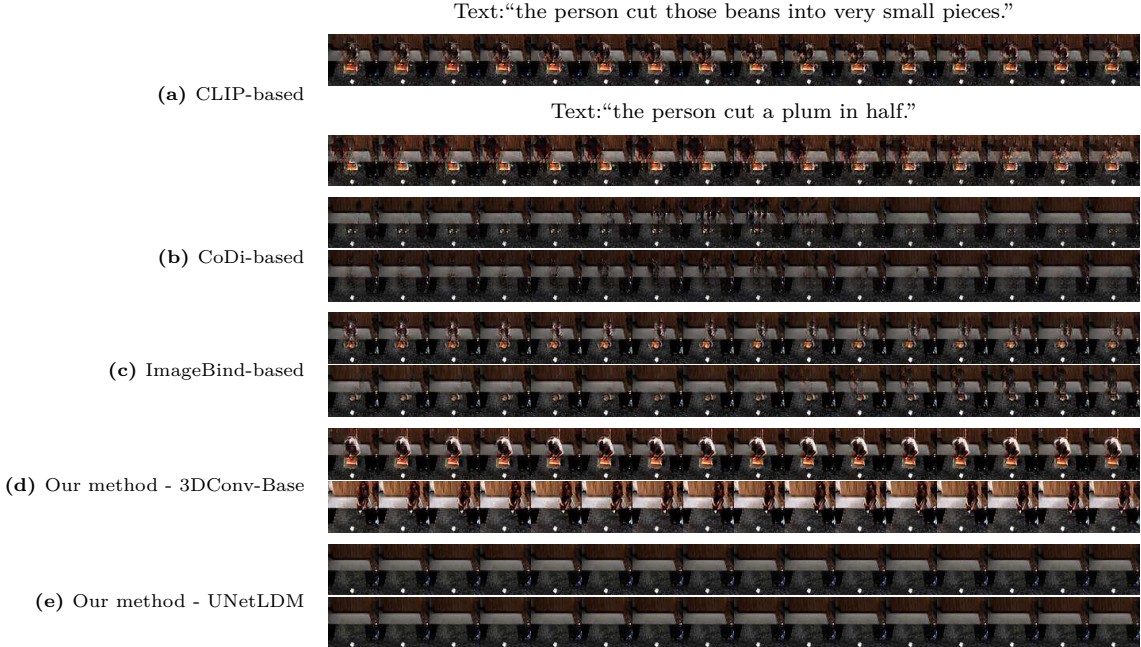

**(a)** CLIP-based

**(b)** CoDi-based

**(c)** ImageBind-based

**(d)** Our method - 3DConv-Base

**(e)** Our method - UNetLDM

**Figure D.8:** Comparison of the generated videos by the alignment models: CLIP-based, CoDi-based, ImageBind-based, our baseline method with both 3DConv-Base and UNetLDM video architectures, on the TACoS data set.

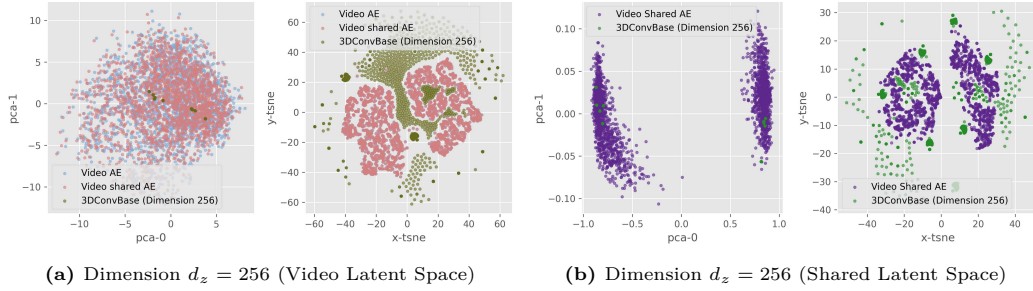

**(a)** Dimension $d_z = 256$ (Video Latent Space)   **(b)** Dimension $d_z = 256$ (Shared Latent Space)

**Figure D.9:** Video latent space (left side) from ablation experiment with video representation learning using $d_z = 256$, showing embeddings from the mapping functions, video shared autoencoders, and video representation learning. Additionally, shared semantic latent space (right side), showing embeddings from the mapping functions and video shared autoencoders, visualized with PCA (odd columns) and t-SNE (even columns).

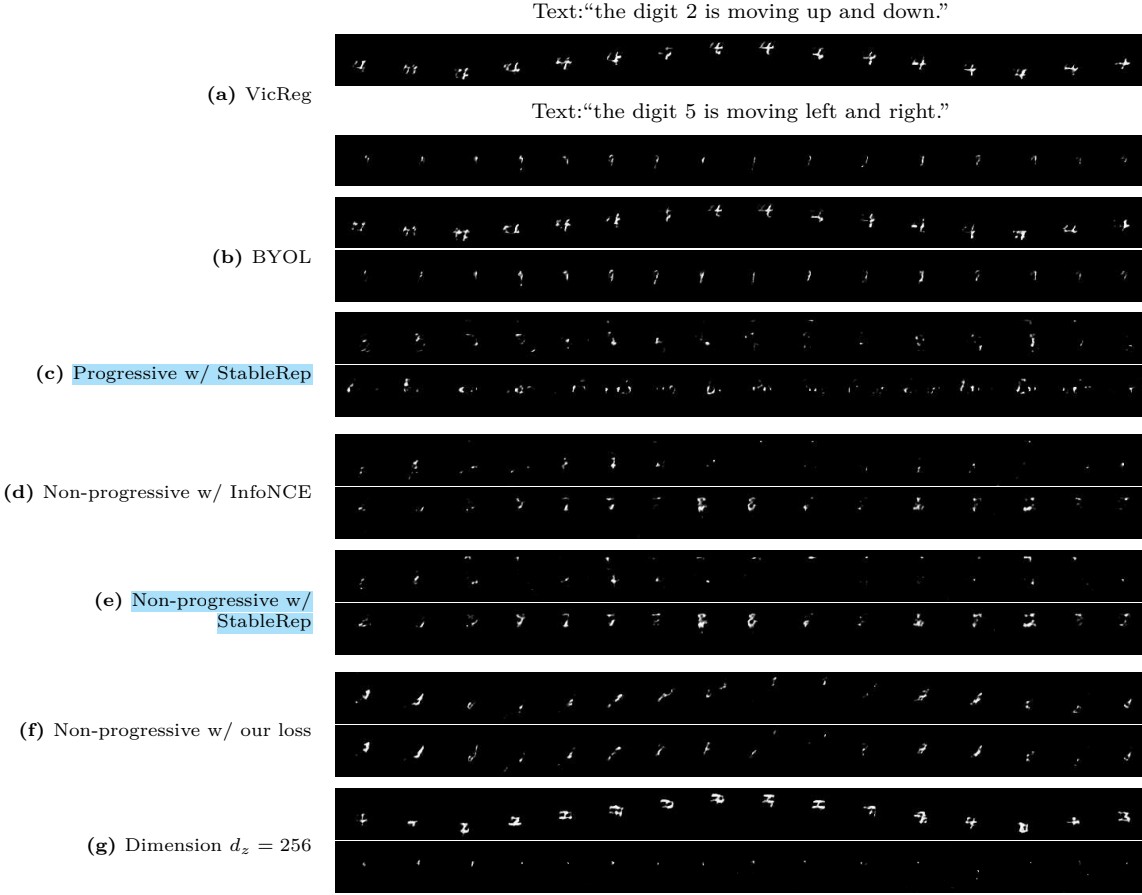

**Figure D.10:** Comparison of the generated videos by the ablation alignment models: VicReg, BYOL, non-progressive with InfoNCE, non-progressive with our baseline loss, and video autoencoder baseline with dimension $d_z = 256$, using the 3DConv-Base video architecture on the SyncDraw-MM data set.

