# OpenReview forum: "On the Analysis of the One-to-Many Mapping in Cross-Modality Text-to-Video Generation with Semantic Spaces"
_TMLR — Under review for TMLR_

### Review · Reviewer_w9t3 · 2026-05-25

**Summary Of Contributions:**

This paper studies the one-to-many mapping problem in text-to-video generation from a representation-learning perspective. The main point is that when one text prompt corresponds to several valid videos, standard feature-alignment objectives may encourage collapse or poor alignment in the shared latent space. To analyze this, the authors propose a progressive text-to-video alignment framework that first maps text into a shared semantic space and then into the video latent space. They also introduce a bucket-based loss, where videos associated with the same text are treated as positives rather than negatives.

The paper evaluates this idea with several video autoencoder backbones and compares against adapted CLIP-, ImageBind-, and CoDi-based alignment baselines on SyncDraw-MM, KTH, TACoS, and a latent-space analysis on Panda70M. The strongest aspect of the paper, in my view, is that it looks beyond final video quality and tries to understand the learned latent spaces directly. The bucket-based evaluation idea is also a reasonable way to acknowledge that a given text does not have a unique ground-truth video.

**Audience:**

Yes

**Audience Explanation:**

The topic is relevant to a subset of the TMLR audience, especially researchers working on multimodal representation learning, video-language learning, and generative modeling. The one-to-many problem is real and important, and the paper's attempt to analyze latent-space behavior rather than only reporting video-quality metrics is interesting.

**Broader Impact Concerns:**

None.

**Claims And Evidence:**

No

**Claims Explanation:**

The paper gives useful evidence that one-to-many text-video alignment is challenging, and I agree that this is an important issue. The experiments also support the narrower observation that video representation choice can strongly affect downstream alignment. However, to my understanding, the current evidence does not yet convincingly establish that the proposed progressive/bucket formulation is a generally effective solution to the problem.

My main concerns are as follows:

1. The baseline comparisons of CLIP, ImageBind, and CoDi are adapted into the authors' framework, but it is not clear how faithful or strong these adapted versions are relative to the original methods. Since these methods were not designed exactly for this setting, the comparison may reflect implementation choices inside the proposed pipeline rather than a clear failure of existing alignment methods.

2. The architecture, loss, target video representation, decoder, and alignment procedure are convoluted. I think it would be important to see a direct comparison where the video representation and architecture are fixed while varying only the loss: InfoNCE, supervised/multi-positive contrastive loss, the proposed bucket loss, and perhaps a probabilistic cross-modal embedding objective.

3. The evaluation protocol needs to be clarified. For datasets such as SyncDraw-MM, where there are very few text buckets, it is important to know whether the test set contains the same text buckets seen during training. If so, the results mainly test generation within seen semantic buckets, not generalization to unseen prompts or unseen semantic combinations.

**Requested Changes:**

1. The paper should explain exactly how CLIP, ImageBind, and CoDi are adapted into the proposed framework, which components are preserved from the original methods, and which components are replaced. As written, it is difficult to know whether the results show a real limitation of these prior approaches or mainly reflect the particular adapted implementations used in this paper.

2. Since the architecture, target video representation, decoder, and alignment procedure all interact, I think an important missing experiment is a direct comparison where the video representation and architecture are fixed and only the alignment loss changes. At minimum, I would like to see InfoNCE, supervised or multi-positive contrastive learning, the proposed bucket loss, and possibly a probabilistic cross-modal embedding objective compared under the same setting.

3. For datasets such as SyncDraw-MM, where there are very few text buckets, it is important to state whether the same captions or semantic buckets appear in both training and testing. If the test set uses seen buckets, then the paper should be careful to describe the evaluation as testing generation within seen semantic categories, rather than generalization to unseen prompts or unseen semantic combinations.

4. Add an evaluation over unseen captions or unseen semantic combinations if possible: this would make the claims substantially stronger. Even a small controlled split on SyncDraw-MM or KTH would help separate interpolation within known buckets from true text-to-video semantic generalization.

5. To my understanding, the proposed loss is closely related to supervised or multi-positive contrastive learning, since several samples associated with the same semantic label/text are treated as positives. Hence, I think authors should discuss the relationship between the proposed bucket loss and existing multi-positive contrastive learning.

6. Discuss relevant missing related work [1-3].

7. Fix typos in the figures and text. For example, Figure 2 contains the typo "Stocastic"; this should be "Stochastic."

References:

[1] Probabilistic Embeddings for Cross-Modal Retrieval

[2] Support-set Bottlenecks for Video-Text Representation Learning

[3] End-to-End Learning of Visual Representations from Uncurated Instructional Videos

---

> ### Author Response · Authors · 2026-06-06
> **Response to Reviewer w9t3**
>
> Dear Reviewer w9t3,
>
> First, we want to thank you for your time and work reviewing this manuscript. Please see our responses below and feel free to point out anything that could further improve our work.
>
> We address the more straightforward questions and comments in this first part of the response, while working on the remaining points that require further investigation and experimentation. We are first addressing item 2 of the main concerns raised, which is also related to points 2 and 5 of the requested changes. We expect to have new results within approximately one week.
>
> We will update the manuscript when more changes are available and once the new requested results are obtained.
>
> > The paper gives useful evidence that one-to-many text-video alignment is challenging, and I agree that this is an important issue. The experiments also support the narrower observation that video representation choice can strongly affect downstream alignment. However, to my understanding, the current evidence does not yet convincingly establish that the proposed progressive/bucket formulation is a generally effective solution to the problem.
>
> Thanks for your constructive comment. We would like to emphasize that our main goal is the characterization and analysis of the one-to-many scenario in the text-to-video generation from a latent space perspective, rather than focusing solely on video quality. Although we propose a progressive strategy and bucket loss to conduct this analysis, we do not claim this solution to be optimal across all aspects of the problem, as each component warrants further investigation, including alternative loss functions, modality-specific architectures, and alignment evaluation strategies, among others.
>
> > The baseline comparisons of CLIP, ImageBind, and CoDi are adapted into the authors' framework, but it is not clear how faithful or strong these adapted versions are relative to the original methods. Since these methods were not designed exactly for this setting, the comparison may reflect implementation choices inside the proposed pipeline rather than a clear failure of existing alignment methods.
>
> We thank the reviewer for raising this concern. As described in the manuscript, we follow only the alignment methods from each model, not their modality-specific representation architectures, which is precisely what the adaptation refers to. For example, for the CLIP-based alignment, we do not use their ViT-based encoder (or even an adaptation) for video, nor do we train the text encoder from scratch. Instead, we adopt their proposed linear projection from each modality to the shared representation space along with the use of InfoNCE loss, following the original alignment proposal from their paper and supplementing with information from the official repository where available.
> But note that this is also the approach of ImageBind for alignment, so the final experiment with CLIP-based method was to use the progressive architecture instead of a linear projection, since the linear projection + InfoNCE loss is already being used by the ImageBind-based approach.
> As another example, CoDi shares each modality-specific representation across the other modalities, acting as shared information during representation learning. However, this introduces modality information leakage across representations, which conflicts with our pure feature-alignment strategy between independent representation spaces.
> The main difference lies in the modality-specific representation learning strategies, which we replace with our proposed architectures (or the CLIP text encoder where applicable). This section has been reviewed to improve clarity in this point.

---

> > ### Author Response · Authors · 2026-06-06
> > **Following the previous response to Reviewer w9t3**
> >
> > > The architecture, loss, target video representation, decoder, and alignment procedure are convoluted. I think it would be important to see a direct comparison where the video representation and architecture are fixed while varying only the loss: InfoNCE, supervised/multi-positive contrastive loss, the proposed bucket loss, and perhaps a probabilistic cross-modal embedding objective.
> >
> > Thanks for pointing this out. Our intention with the ablation section was to provide that kind of information with a fixed data set (SyncDraw-MM) and a fixed architecture (3DConv-Base). The results on Section 4.4 also have some variations where some parts are fixed to have complementary information. For instance, we have a linear projection + InfoNCE loss with 3DConv-Base video architecture (ImageBind-based) and progressive architecture + InfoNCE loss with 3DConv-Base video architecture (CLIP-based) for all sets.
> > At the same time, we have the linear projection + UNetLDM architecture (CoDi-based) and progressive architecture + UNetLDM architecture (UNetLDM) for all sets as well. The only experiment not included was linear projection + bucket loss, since the results with all linear projection-based methods did not perform well in alignment, not being able to properly align not even the video shared autoencoder embeddings with the video autoencoder ones (Section 4.4).
> >
> > For the suggestions of including new losses in the experiments, we are working on adding these new results in the revised manuscript, as they require new implementation effort and additional experimentation time.
> >
> > > The evaluation protocol needs to be clarified. For datasets such as SyncDraw-MM, where there are very few text buckets, it is important to know whether the test set contains the same text buckets seen during training. If so, the results mainly test generation within seen semantic buckets, not generalization to unseen prompts or unseen semantic combinations.
> >
> > Thanks for raising this concern. This is a good point. We do not share the training buckets with the testing buckets or vice-versa, i.e., the same bucket is not present in both sets. But the buckets can share the same movement between them. For example, SyncDraw-MM will have in the training set the bucket "the digit 2 is moving left and right'' but not in the test set. In the test set, we will have the same movement "the digit X is moving left and right'' with other digits that were not present in training (``the digit 4 is moving left and right'' is available in testing that is not in training).
> > As another example, in the KTH set, we will have an identified person X on the training split but not the same person with the same movement in the testing split. For example, "person 16 is running left to right'' is available in test set, but the "person 16'' is not present in the training set with any movement, but there are other persons with the same movement available in training, such as "person 22 is running left to right'' (id not available in testing). This section has been reviewed to improve clarity in this point.
> >
> > > 7. Fix typos in the figures and text. For example, Figure 2 contains the typo "Stocastic"; this should be "Stochastic."
> >
> > Thank you for pointing this. We have reviewed the manuscript to address such cases and to detect other typos and textual problems. Please see the revised manuscript with the corrections

---

> > > ### Author Response · Authors · 2026-06-24
> > > **Following the previous response to Reviewer w9t3**
> > >
> > > We have finished the multi-positive investigation and experiments, for which we continue the responses here.
> > >
> > > > The architecture, loss, target video representation, decoder, and alignment procedure are convoluted. I think it would be important to see a direct comparison where the video representation and architecture are fixed while varying only the loss: InfoNCE, supervised/multi-positive contrastive loss, the proposed bucket loss, and perhaps a probabilistic cross-modal embedding objective.
> > >
> > > Regarding the losses: the experiments with other multi-positive losses are also discussed in the last following point, so we leave the main discussion for the additional related works in this item.
> > >
> > > > 5. To my understanding, the proposed loss is closely related to supervised or multi-positive contrastive learning, since several samples associated with the same semantic label/text are treated as positives. Hence, I think authors should discuss the relationship between the proposed bucket loss and existing multi-positive contrastive learning.
> > >
> > > Thanks for the feedback. This point is also related to the next point for which we add more details and discussion.
> > >
> > > > Discuss relevant missing related work [1-3].
> > > References:
> > > [1] Probabilistic Embeddings for Cross-Modal Retrieval
> > > [2] Support-set Bottlenecks for Video-Text Representation Learning
> > > [3] End-to-End Learning of Visual Representations from Uncurated Instructional Videos
> > >
> > > Thanks for pointing these related works. We have checked that the last given reference ([3]) is more directly related to a multi-positive setup, while [1] is a probabilistic-based setup considered as a partially/soft multi-positive loss with a different paradigm and [2] is not a multi-positive loss.
> > >
> > > We started by considering the [3]-based loss for comparison, but found that their setup is different from our batch formulation which adds a different component in the comparison, which is not our goal to evaluate. The MIL-NCE [3] loss consider a different batch formulation, where the batch has a maximum of N positive samples for each instance. So, for our context with text, each text will have at maximum N positive associated samples. If the text does not have such samples, the existing positives samples are repeated. Moreover, batch size and the candidate number (positive samples) are obtained from the best results of their ablation experiments in their context. So they are tied to the problem itself, which is the relation of a video being described by several different narrations (texts). We do not restrict our batch formulation to consider a minimum or maximum number of positive samples or negative samples, and neither setup a specific batch formulation for training.
> > >
> > > Regarding [1] used for classification retrieval task related with image-text in a one-to-many setting, their loss does not model a explicit positive set, but rather a different paradigm for each sample as a distribution from which a Monte Carlo sampling is applied. Their family loss is pair-based in this setup instead of a (positive) set-based loss. Nevertheless, do note that this loss requires extra regularization terms. Their setup does not model directly a multi-positive setup but rather is a side effect of the probabilistic formulation, which represents a different mechanism. Future work could consider this family to understand the adaptation required for training and inference, which follows a different protocol, and other similar family losses for comparison.
> > >
> > > Regarding related work [2] also for retrieval tasks, an additional task related to text generation is considered in the loss with a standard noise contrastive loss (NCE). So, NCE is the primary discriminative objective handling single-positive scenario used with a generative cross-instance captioning loss (obtained from a text generation module). So, is not an adaptation to the loss itself to deal with multi-positive but rather they add an auxiliary generative loss outside the contrastive loss. The mechanisms are different and a comparison ends up considering contrastive learning quality with generative modeling quality.
> > >
> > > Moreover, we also considered other related multi-positive losses such as the SupCon [4], which first seemed to be directly related. But their formulation require two augmented versions of each sample, where these augmentation are performed by external works/models that they train, having only one option which deals with simpler augmentation techniques for images.
> > > Nevertheless the official paper implementation directs to another related multi-positive loss called StableRep [5]. This was the more direct loss that does not assume different batch formulation, consider a variable number of positive samples for each instance (or text), and does not require external modules to train to compose the loss that deal with a multi-positive setting.

---

> > > > ### Author Response · Authors · 2026-06-24
> > > > **Following the previous response to Reviewer w9t3**
> > > >
> > > > We added a brief discussion in the manuscript related to these works as well the results of StableRep loss in our context.
> > > >
> > > > [4] Supervised Contrastive Learning, In NeurIPS, 2020.
> > > > [5] StableRep: Synthetic Images from Text-to-Image Models Make Strong Visual Representation Learners, NeurIPS, 2023.

---

> > > > > ### Author Response · Authors · 2026-06-27
> > > > > **Following the previous response to Reviewer w9t3 with updates**
> > > > >
> > > > > Dear Reviewer w9t3,
> > > > >
> > > > > We have included a quantitative alignment analysis in the revised manuscript, prompting an updated discussion of the results, including those with StableRep — an additional multi-positive contrastive loss included as requested. We invite the reviewer to consult the revised manuscript.

---

### Review · Reviewer_L7pR · 2026-06-02

**Summary Of Contributions:**

This paper studies the one-to-many mapping problem in feature-alignment approaches to text-to-video generation. The authors argue that, when a single text description can correspond to many valid videos, direct cross-modal feature alignment can lead to representation collapse or poor alignment. To analyze this problem, the paper builds a plug-and-play pipeline with unsupervised video representation learning, a CLIP text encoder, and a progressive text-to-video latent alignment model. The proposed alignment uses an intermediate shared semantic space and a bucket-based loss that treats videos sharing the same text description as positives. The paper evaluates this setup on SyncDraw-MM, KTH, TACoS, and, for latent-space characterization, Panda70M. It compares several video autoencoder architectures and adapted CLIP/ImageBind/CoDi-style alignment baselines, using bucket-based full-reference metrics, distributional video metrics, and PCA/t-SNE visualizations of latent spaces.

The main strength of the paper is that it focuses on a real and under-discussed issue in text-to-video representation alignment: the ambiguity induced by multiple valid video realizations for the same text. The paper is also valuable in separating video representation learning from cross-modal alignment and in showing that video representation choice can substantially affect downstream alignment. The experiments are fairly broad for an analysis paper and include multiple datasets, several alignment variants, ablations, qualitative examples, and implementation details.

The main weakness is that several central conclusions are stronger than the evidence currently supports. Many claims about representation collapse, alignment quality, and the superiority of progressive/bucket-based alignment rely heavily on 2D PCA/t-SNE plots and qualitative interpretation, while the quantitative metrics often show small or mixed differences across methods. In addition, the baselines are adapted rather than used in their original intended form, the semantic quality of generated videos is not directly evaluated by humans or a text-video consistency metric, and the promised code/checkpoints link is currently a placeholder. These issues make the paper interesting but not yet fully convincing under TMLR's evidence criterion.

**Audience:**

Yes

**Audience Explanation:**

Yes. The topic fits TMLR and should interest readers working on multimodal representation learning, text-to-video generation, and generative-model evaluation. The central observation, that one-to-many mappings can make feature-level alignment behave differently from one-to-one alignment, is relevant even if the method is not a new state of the art generator.

**Broader Impact Concerns:**

I do not see a severe unaddressed ethical issue.

**Claims And Evidence:**

No

**Claims Explanation:**

The paper provides useful experiments, but the evidence is not yet convincing enough for the main claims as stated. The strongest claims about representation collapse and better alignment are supported mainly by visual inspection of PCA/t-SNE plots. These are helpful diagnostics, but they should be backed by quantitative measures in the original latent space, such as intra-/inter-bucket separation, generated-code coverage, nearest-neighbor bucket purity, diversity, or distance to the support of real video codes.

The quantitative tables also do not always cleanly support the narrative. Different methods win on different metrics and datasets, and the authors note that several alignment methods have similar quantitative results. I would like the paper to tie each major finding more directly to numerical evidence, and soften claims where the support is primarily qualitative.

**Requested Changes:**

- Add quantitative evidence for collapse/alignment in the original latent spaces, rather than relying mainly on PCA/t-SNE plots.

- Add direct semantic evaluation of text-video consistency, e.g. simple attribute probes for SyncDraw-MM/KTH or a small human/text-video consistency evaluation.

---

> ### Author Response · Authors · 2026-06-24
> **Response to Reviewer L7pR**
>
> Dear Reviewer L7pR,
>
> First, we want to thank you for your time and work reviewing this manuscript. Please see our responses below and feel free to point out anything that could further improve our work.
>
> We have generated metrics for the quantitative evaluation of the alignment and we are working on the inclusion of these results in a newer version of the manuscript. We expect to finish this inclusion in the next 3 days.
> Meanwhile, we address the other questions and comments in this first part of the response.
>
> > The main weakness is that several central conclusions are stronger than the evidence currently supports. Many claims about representation collapse, alignment quality, and the superiority of progressive/bucket-based alignment rely heavily on 2D PCA/t-SNE plots and qualitative interpretation, while the quantitative metrics often show small or mixed differences across methods. In addition, the baselines are adapted rather than used in their original intended form, the semantic quality of generated videos is not directly evaluated by humans or a text-video consistency metric, and the promised code/checkpoints link is currently a placeholder. These issues make the paper interesting but not yet fully convincing under TMLR's evidence criterion.
>
> Thanks for your constructive comments and feedback. We have reviewed the manuscript following the below points, please check the newer version.
>
> Regarding the adapted versions, concern also raised by Reviewer w9t3, we only follow the alignment methods from each model, not their modality-specific representation architectures as these models are not directly applied to our scenario as they are defined originally. Please check the newer version of the manuscript where we better clarify this and the answer for the Reviewer w9t3 which raised a similar concern in their first point.
>
> The code and checkpoints will be released under the manuscript approval. We updated the footnote to be more transparent.
>
> > The quantitative tables also do not always cleanly support the narrative. Different methods win on different metrics and datasets, and the authors note that several alignment methods have similar quantitative results. I would like the paper to tie each major finding more directly to numerical evidence, and soften claims where the support is primarily qualitative.
>
> We thank the reviewer for raising this point. The quantitative tables address video quality assessment, while the latent space analysis serves as a complementary assessment of alignment. We have reviewed the claims in Section 4 related to these two results to be sure they are more tied with them. Please check the revised manuscript and feel free to point out specific claims that may still need to be revised or softened.
>
> Note: the changes have been highlighted throughout the manuscript; however, highlights nested within other highlighted sections may not render properly. Therefore, please also review the revised claims in ``Finding 3.''
>
> > Add direct semantic evaluation of text-video consistency, e.g. simple attribute probes for SyncDraw-MM/KTH or a small human/text-video consistency evaluation.
>
> We thank the reviewer for this suggestion and we agree that it would be complementary to include such evaluation, specifically the text-video consistency metric. However, human-based evaluation is not an easy, dynamic, or fast approach within our institution, as ethics committees are typically involved in approving such evaluations and, based on previous experience, this process can take several months.
>
> Regardless, given that we will include the alignment quantitative evaluation, we found these results sufficient in the manuscript context to address the alignment problem, which remains our primary focus in this manuscript. Future work could explore additional metrics and evaluation strategies beyond alignment and video quality.

---

> ### Author Response · Authors · 2026-06-27
> **Following the previous response to Reviewer L7pR with updates**
>
> Dear Reviewer L7pR, please find below our updates addressing the requested points.
>
> > I would like the paper to tie each major finding more directly to numerical evidence, and soften claims where the support is primarily qualitative.
>
> We noticed that the changes to "Finding 3" did not render correctly in the previous submission. Please refer to the latest version of the manuscript, in which this has been corrected.
>
> > Add quantitative evidence for collapse/alignment in the original latent spaces, rather than relying mainly on PCA/t-SNE plots.
>
> Thanks for your constructive suggestions. We have added the Chamfer and Hausdorff distances, computed in the original latent spaces, along with the uniformity metric [1], as quantitative alignment metrics to complement the latent space analysis. The discussion has been revised accordingly to incorporate these new results.
>
> [1] Wang and Isola. Understanding Contrastive Representation Learning through Alignment and Uniformity on the Hypersphere. In ICML, 2020

---

### Review · Reviewer_JSQC · 2026-06-09

**Summary Of Contributions:**

This paper studies the one-to-many mapping problem in cross-modality text-to-video generation. The basic point is that a single text prompt can correspond to many valid videos, but standard alignment losses often behave as if there should be only one correct match.

To study this issue, the paper proposes a unidirectional progressive learning framework. Instead of directly mapping text to the video latent space, the method first maps text into a shared semantic space and then maps that representation into a pre-trained video latent space. The authors also introduce a Bucket Loss, where videos sharing the same text caption are grouped into a bucket and treated as positive samples. The experiments focus on latent space analysis using PCA, t-SNE, and UMAP, along with standard video generation metrics on datasets such as SyncDraw-MM, KTH, and TACoS.

**Audience:**

Yes

**Audience Explanation:**

Despite the empirical weaknesses, the topic is important. The paper highlights a real problem in multimodal alignment: standard contrastive losses can punish valid alternative outputs when one input has many correct targets. This matters for text-to-video generation, text-to-image generation, and more general multimodal representation learning.

The architectural analysis is also interesting. The finding that different video autoencoders produce latent spaces with very different alignment behavior could be useful for researchers designing cross-modal generative models. Even if the proposed method is not yet convincing as a final solution, the paper’s diagnosis of the problem is worth paying attention to.

**Broader Impact Concerns:**

The paper focuses on a foundational representation learning problem, and the datasets used in the experiments are mostly standard and low-risk. Still, the bucket idea has a potential bias issue.

If a model maps a generic prompt such as “a person running” to a bucket of videos, then the diversity of that bucket matters a lot. If the bucket mostly contains people from one demographic group, one camera style, or one type of motion, the model may learn that this narrow version is the default interpretation of the prompt. In that sense, the one-to-many mapping could still collapse toward the majority pattern inside the bucket.

The paper should include a short broader impact discussion about this. In particular, it should acknowledge that bucket construction is data-dependent, and that biased or unbalanced buckets could lead to biased generations.

**Claims And Evidence:**

No

**Claims Explanation:**

The paper presents a lot of qualitative evidence through latent space plots, but the evidence is not strong enough to fully support the main claims.

First, the generation quality is weak. The paper claims that the progressive architecture favors alignment compared to projection layers, but the quantitative results do not show a decisive advantage. In Table 2, the bucket-based PSNR and SSIM numbers are often close across methods. Some improvements are visible, but they are not large enough to establish a clear win. More importantly, the FVD scores remain high in several settings, and the UNetLDM version performs especially poorly on TACoS. That is a serious issue, because UNetLDM is supposed to be the stronger reconstructive model. If the proposed alignment framework breaks down for that representation, the limitation needs to be treated as central rather than incidental.

Second, the latent space evidence is suggestive but not conclusive. t-SNE and UMAP can reveal useful local structure, but they also distort distances and global geometry. The paper sometimes makes strong statements about generated distributions falling outside expected regions or about certain methods producing collapse. These may be true, but the plots alone do not prove it. The paper needs high-dimensional quantitative measures, such as latent Fréchet distance, alignment and uniformity metrics, nearest-neighbor statistics, or other distributional comparisons.

Third, the Bucket Loss is evaluated only in a constrained version of the one-to-many problem. The buckets are built from identical text strings. This does not really test the harder natural-language case, where semantic equivalence is not the same as exact caption equality. As a result, the paper shows that the method can exploit dataset-defined repeated captions, but it does not show that it solves one-to-many semantic mapping in realistic text-to-video data.

**Requested Changes:**

Current data set used in the evaluation is a super toy. So I would strongly suggest the author to include the larger and more complex data set in the evaluations.

The authors should add quantitative latent alignment metrics. The current 2D visualizations should be supplemented with measurements in the original latent space. Possible options include Fréchet distance between latent distributions, alignment and uniformity metrics, nearest-neighbor overlap, distribution coverage, or mutual-information-style measures. The paper’s visual claims need stronger numerical support.

The authors should address the exact-string limitation of Bucket Loss. At minimum, the paper should clearly state this as a limitation. Ideally, the authors should test a semantic bucketing strategy, for example by grouping captions using a pre-trained language model and a cosine similarity threshold. This would make the method much more relevant to real text-to-video data.

The authors need to explain the UNetLDM failure more carefully. UNetLDM appears to be a better reconstruction model, but it becomes harder to align and even collapses badly on TACoS. This is not a minor detail. If the progressive mapping is incompatible with sparse or highly structured latent spaces, then the proposed framework has a serious limitation.

---

> ### Author Response · Authors · 2026-06-24
> **Response to reviewer JSQC**
>
> Dear Reviewer JSQC,
>
> First, we want to thank you for your time and work reviewing this manuscript. Please see our first part of the responses below and feel free to point out anything that could further improve our work.
>
> We have generated quantitative results for the alignment to support the latent space analysis. We are currently working on the inclusion of these results in a newer version of the manuscript. We expect to finish this inclusion in the next 3 days. Meanwhile, we address other questions and comments in this first part of the response.
>
> > Current data set used in the evaluation is a super toy. So I would strongly suggest the author to include the larger and more complex data set in the evaluations.
>
> Thanks for you suggestion. We agree that SyncDraw-MM and KTH are toy datasets for video generation and fusion-based text-to-video generation. But not for pure-alignment text-to-video generation, as for these sets this is not solved yet. We add a medium size dataset which is the TACoS set with 49k video clips to further evaluate different complexities and sizes of the one-to-many scenario and Panda70M test set (100k) to characterize the problem.
> Our main goal in this manuscript is the characterization and analysis of the one-to-many scenario in a pure-alignment text-to-video generation, so the analysis of increasing complexity is important. Future work could focus on larger sets only, noting that these smaller sets could be a subset (problem) of a larger set.
>
> > The authors should address the exact-string limitation of Bucket Loss. At minimum, the paper should clearly state this as a limitation. Ideally, the authors should test a semantic bucketing strategy, for example by grouping captions using a pre-trained language model and a cosine similarity threshold. This would make the method much more relevant to real text-to-video data.
>
> Thanks for raising this concern. With your comment we identified that we were not clear about the mask building setup. In the loss definition of Equation 4, we define as using the same text embedding to start building such mask but we did not give more details on how we define the sample belonging to the same bucket or not. We do not use the exact same string (or representation) to consider the match, although this can be used as well, but rather implement a cosine similarity between them with a threshold (e.g., 0.999) to identify semantically similar or not similar texts.
> Please check the reviewed manuscript version with this update.
>
> > The authors need to explain the UNetLDM failure more carefully. UNetLDM appears to be a better reconstruction model, but it becomes harder to align and even collapses badly on TACoS. This is not a minor detail. If the progressive mapping is incompatible with sparse or highly structured latent spaces, then the proposed framework has a serious limitation.
>
> We thank the reviewer for raising this concern. The result with UNetLDM, mostly the collapse with TACoS set, does not necessarily indicate a limitation of the progressive approach with sparser and highly structured latent spaces, but rather a limitation of the UNetLDM representational space with our baseline progressive mapping architecture, as this model presents a different result with a linear projection architecture in a progressive mapping (mapping to a shared space and then to a video semantic space).
> To identify this limitation, other video autoencoder architectures that produce sparser and highly structured latent spaces needed to be evaluated.
> We also note that in the ablation experiments, we made a tentative on evaluating different latent space dimensions of SyncDraw-MM to explore a sparser setup for this scenario.
>
> We have reviewed the manuscript to improve clarity on this point and also noted that the term "progressive mapping" was being used too broadly, which may have mislead the reader into confusing a progressive architecture (the mapping function architecture) with a progressive approach (the two-stage mapping to a shared semantic space and then to the video space).

---

> > ### Author Response · Authors · 2026-06-27
> > **Following the previous response to Reviewer JSQC with updates**
> >
> > Dear reviewer JSQC, please find below our updates addressing the requested points.
> >
> > > The authors should add quantitative latent alignment metrics. The current 2D visualizations should be supplemented with measurements in the original latent space. Possible options include Fréchet distance between latent distributions, alignment and uniformity metrics, nearest-neighbor overlap, distribution coverage, or mutual-information-style measures. The paper’s visual claims need stronger numerical support.
> >
> > Thanks for your constructive suggestions. We have added the Chamfer and Hausdorff distances, computed in the original latent spaces, along with the uniformity metric [1], as quantitative alignment metrics to complement the latent space analysis. The discussion has been revised accordingly to incorporate these new results.
> >
> > [1] Wang and Isola. Understanding Contrastive Representation Learning through Alignment and Uniformity on the Hypersphere. In ICML, 2020

---

> > > ### Comment · Reviewer_JSQC · 2026-07-04
> > >
> > > Minor revisions needed: Please adjust the color scheme used in Figure 3. All data points share the same color and heavily overlap, resulting in a solid color block with no distinguishable individual dots.

---

> > ### Comment · Reviewer_JSQC · 2026-07-04
> >
> > For Anwser 1: what exactly does the term "pure-alignment text-to-video generation" refer to? Text-driven video generation inherently falls under the umbrella of text-to-video generation; these are not distinct research tasks. Furthermore, the paper does not introduce a standalone research problem labelled "pure alignment text-to-video generation". It only brings a missing perspective here. Given this reasoning, there is no justification to isolate the dataset used, nor any valid reason for omitting comparisons against this dataset.
> >
> > For Answer 2. Thanks for the clarification. I now get the point.